# Enhancing mitochondrial activity in neurons protects against neurodegeneration in a mouse model of multiple sclerosis

Sina C Rosenkranz[1], Artem A Shaposhnykov[1], Simone Träger[1], Jan Broder Engler[1], Maarten E Witte[2,3], Vanessa Roth[1], Vanessa Vieira[1], Nanne Paauw[3], Simone Bauer[1], Celina Schwencke-Westphal[1], Charlotte Schubert[1], Lukas Can Bal[1], Benjamin Schattling[1], Ole Pless[4], Jack van Horssen[3], Marc Freichel[5], Manuel A Friese[1]*

[1]Institute of Neuroimmunology and Multiple Sclerosis (INIMS), University Medical Center Hamburg-Eppendorf, Hamburg, Germany; [2]Department of Pathology, Amsterdam UMC, MS Center Amsterdam, Amsterdam Neuroscience, Amsterdam, Netherlands; [3]Department of Molecular Cell Biology and Immunology, Amsterdam UMC, MS Center Amsterdam, Amsterdam Neuroscience, Amsterdam, Netherlands; [4]Fraunhofer ITMP ScreeningPort, Hamburg, Germany; [5]Institute of Pharmacology, Heidelberg University, Heidelberg, Germany

**Abstract** While transcripts of neuronal mitochondrial genes are strongly suppressed in central nervous system inflammation, it is unknown whether this results in mitochondrial dysfunction and whether an increase of mitochondrial function can rescue neurodegeneration. Here, we show that predominantly genes of the electron transport chain are suppressed in inflamed mouse neurons, resulting in impaired mitochondrial complex IV activity. This was associated with post-translational inactivation of the transcriptional co-regulator proliferator-activated receptor gamma coactivator 1-alpha (PGC-1$\alpha$). In mice, neuronal overexpression of *Ppargc1a*, which encodes for PGC-1$\alpha$, led to increased numbers of mitochondria, complex IV activity, and maximum respiratory capacity. Moreover, *Ppargc1a*-overexpressing neurons showed a higher mitochondrial membrane potential that related to an improved calcium buffering capacity. Accordingly, neuronal deletion of *Ppargc1a* aggravated neurodegeneration during experimental autoimmune encephalomyelitis, while neuronal overexpression of *Ppargc1a* ameliorated it. Our study provides systemic insights into mitochondrial dysfunction in neurons during inflammation and commends elevation of mitochondrial activity as a promising neuroprotective strategy.

*For correspondence:
manuel.friese@zmnh.uni-hamburg.de

Competing interests: The authors declare that no competing interests exist.

## Introduction

Multiple sclerosis (MS) is a chronic inflammatory disease of the central nervous system (CNS) and the most frequent non-traumatic cause of neurological impairment during early adulthood. Neuronal loss occurs already from disease onset and correlates best with irreversible disability in MS patients (*Fisher et al., 2008*; *Fisniku et al., 2008*; *Tallantyre et al., 2010*). While there has been substantial progress in the understanding and treatment of the immune response, the pathogenesis of concurrent neuronal damage is incompletely understood. Recently, two drugs, ocrelizumab and siponimod, have been licensed for the treatment of progressive MS patients; however, their mode of action relies on immune regulation (*Kappos et al., 2018*; *Montalban et al., 2017*). Since there are currently no therapies available to counteract the progression of neurodegeneration in MS patients

**eLife digest** Multiple sclerosis is a life-long neurological condition that typically begins when people are in their twenties or thirties. Symptoms vary between individuals, and within a single individual over time, but can include difficulties with vision, balance, movement and thinking. These occur because the immune system of people with multiple sclerosis attacks the brain and spinal cord. This immune assault damages neurons and can eventually cause them to die. But exactly how this happens is unclear, and there are no drugs available that can prevent it.

One idea is that the immune attack in multiple sclerosis damages neurons by disrupting structures inside them called mitochondria. These cellular 'organs', or organelles, produce the energy that all cells need to function correctly. If the mitochondria fail to generate enough energy, the cells can die. And because neurons are very active cells with high energy demands, they are particularly vulnerable to the effects of mitochondrial damage.

By studying a mouse version of multiple sclerosis, Rosenkranz et al. now show that mitochondria in the neurons of affected animals are less active than those of healthy control mice. This is because the genes inside mitochondria that enable the organelles to produce energy are less active in the multiple sclerosis mice. Most of these genes that determine mitochondrial activity and energy production are under the control of a single master gene called PGC-1alpha. Rosenkranz et al. showed that boosting the activity of this gene — by introducing extra copies of it into neurons — increases mitochondrial activity in mice. It also makes the animals more resistant to the effects of multiple sclerosis.

Boosting the activity of mitochondria in neurons could thus be a worthwhile therapeutic strategy to investigate for multiple sclerosis. Future studies should examine whether drugs that activate PGC-1alpha, for example, could help prevent neuronal death and the resulting symptoms of multiple sclerosis.

(*Dendrou et al., 2015*; *Feinstein et al., 2015*) and repurposing of drugs to directly target neurodegeneration in MS has been disappointing (*Chataway et al., 2020*), a better understanding of the molecular processes that determine neuronal cell loss in MS is urgently needed.

Neuronal loss in MS and its animal model, experimental autoimmune encephalomyelitis (EAE), has been associated with enhanced production of reactive oxygen and nitrogen species by immune cells and with increased iron accumulation in the gray matter (*Stephenson et al., 2014*). Both processes can lead to damage of neuronal mitochondria with subsequent metabolic failure (*Campbell et al., 2011*). Moreover, disruption of neuronal ion homeostasis (*Craner et al., 2004*; *Friese et al., 2014*) and aggregation of neuronal proteins (*Schattling et al., 2019*) consume high amounts of energy that might further drive neuroaxonal injury. Furthermore, excessive activation of calcium-dependent processes and neuronal calcium overload seems to be another important component of neuronal injury (*Friese et al., 2007*; *Schattling et al., 2012*; *Witte et al., 2019*). While identifying druggable targets that specifically induce neuronal resilience has been extremely difficult due to insufficient insights into key modulators, mitochondria could serve as an important hub as they are pivotal for both energy production and calcium homeostasis (*Vafai and Mootha, 2012*).

Mitochondria usually have a high calcium buffering capacity, which is driven by their negative membrane potential that is generated by the activity of oxidative phosphorylation (*Kann and Kovács, 2007*; *Zorova et al., 2018*). However, overload of mitochondria with calcium as a consequence of CNS inflammation can result in inappropriate activation of the mitochondrial permeability transition pore (PTP) with subsequent mitochondrial swelling and cell death (*Giorgi et al., 2018*; *Rizzuto et al., 2012*), which is one of the neuropathological hallmarks in neurons during EAE (*Nikić et al., 2011*). Inhibiting the mitochondrial matrix protein cyclophilin D, a regulator of the PTP can partially counteract this dysregulation (*Forte et al., 2007*). Additionally, to counteract ion imbalance, a higher activity of ATP-dependent ion pumps would be required. However, postmortem studies of MS patients' CNS tissue revealed a compromised neuronal ATP production by oxidative phosphorylation as decreased mitochondria complex IV activity in demyelinated axons and neurons was detected (*Campbell et al., 2011*; *Mahad et al., 2009*). Similarly, mitochondrial gene expression

is suppressed in motor neurons of the spinal cord in EAE mice (*Schattling et al., 2019*), further supporting a comprised mitochondrial function during CNS inflammation.

Thus, excess neuroaxonal calcium, together with mitochondrial dysfunction, has been postulated to trigger neuroaxonal injury observed in MS and EAE. This would be predicted to lead to elevated levels of calcium within neuronal and axonal mitochondria, further perpetuating mitochondrial dysfunction and neuronal injury. However, it is currently unknown whether interventions that increase mitochondrial energy production in neurons can rescue neurodegeneration during CNS inflammation.

Here, we discovered by an unsupervised survey of neuronal gene expression during CNS inflammation that genes involved in the electron transport chain (ETC), especially in complex I and IV, are repressed in motor neurons. While this is not accompanied by a decrease of mitochondrial numbers in motor neuronal somata, we detected a decrease in neuronal mitochondrial complex IV activity. This implies a lower activity of neuronal mitochondria during EAE, and we were able to associate that with a post-translational inactivation of peroxisome proliferator-activated receptor gamma coactivator 1-alpha (PGC-1α), one of the master regulators of mitochondrial numbers and function. Notably, neuronal overexpression of *Ppargc1a*, which encodes for PGC-1α, led to an increase of mitochondrial activity, especially in complex IV activity and substantially elevated calcium buffering capacity. Subjecting these mice to EAE resulted in a significantly better recovery from clinical disability in comparison to wild-type controls. Together, induction of neuronal mitochondrial activity commends as a promising therapeutic approach to counteract inflammation-induced neurodegeneration.

## Results

### Impaired neuronal oxidative phosphorylation in inflamed spinal cords

Since we have previously discovered that mitochondrial gene transcripts are markedly suppressed in motor neurons during CNS inflammation in the EAE model (*Schattling et al., 2019*), we first investigated whether unifying biological processes are dysregulated. Therefore, we derived a gene expression signature of motor neurons in spinal cord during CNS inflammation (*Schattling et al., 2019*) and tested for gene set enrichment of biological process gene ontology (GO) terms. While the majority of terms was enriched during inflammation, only a confined group of 29 terms showed strong de-enrichment (*Figure 1A*, dashed line) and was dominated by terms representing mitochondrial function (*Figure 1B*). A recurring theme was the suppression of the ETC (*Figure 1C*). Suppression of gene transcripts that are involved in the ETC, which is the key mechanism of oxidative phosphorylation, can either result in reduced number of mitochondria or an impairment of mitochondrial function. Therefore, we quantified mitochondria content in the soma of motor neurons in EAE mice at the chronic stage in comparison to healthy control mice, but did not detect any difference in mitochondria area normalized to neuronal size (*Figure 1D*, *Figure 1—figure supplement 1A*). However, we detected a tendency towards an increase in overall numbers of motor neuronal mitochondria (*Figure 1—figure supplement 1B*) that was based on an increased size of the neuronal cell body during EAE (*Figure 1E*). By contrast, the size of mitochondria did not differ in motor neurons of healthy and EAE mice (*Figure 1—figure supplement 1C*).

Dismissing differences in mitochondrial content, we reasoned that downregulation of ETC transcripts could result in a compromised oxidative phosphorylation activity. By analyzing COX histochemistry that represents complex IV activity, we detected a significant decrease in absolute neuronal complex IV activity in the entire gray matter and ventral horn of acute EAE animals (day 15 post immunization) and in chronic EAE animals (day 30 post immunization) in comparison to healthy control mice (*Figure 1F*). Normalization to HuC/HuD-positive neurons revealed that the reduced complex IV activity at day 30 was partly driven by neuronal loss, whereas at day 15 the reduced complex IV activity was independent of neuronal numbers (*Figure 1G*, *Figure 1—figure supplement 1D–E*).

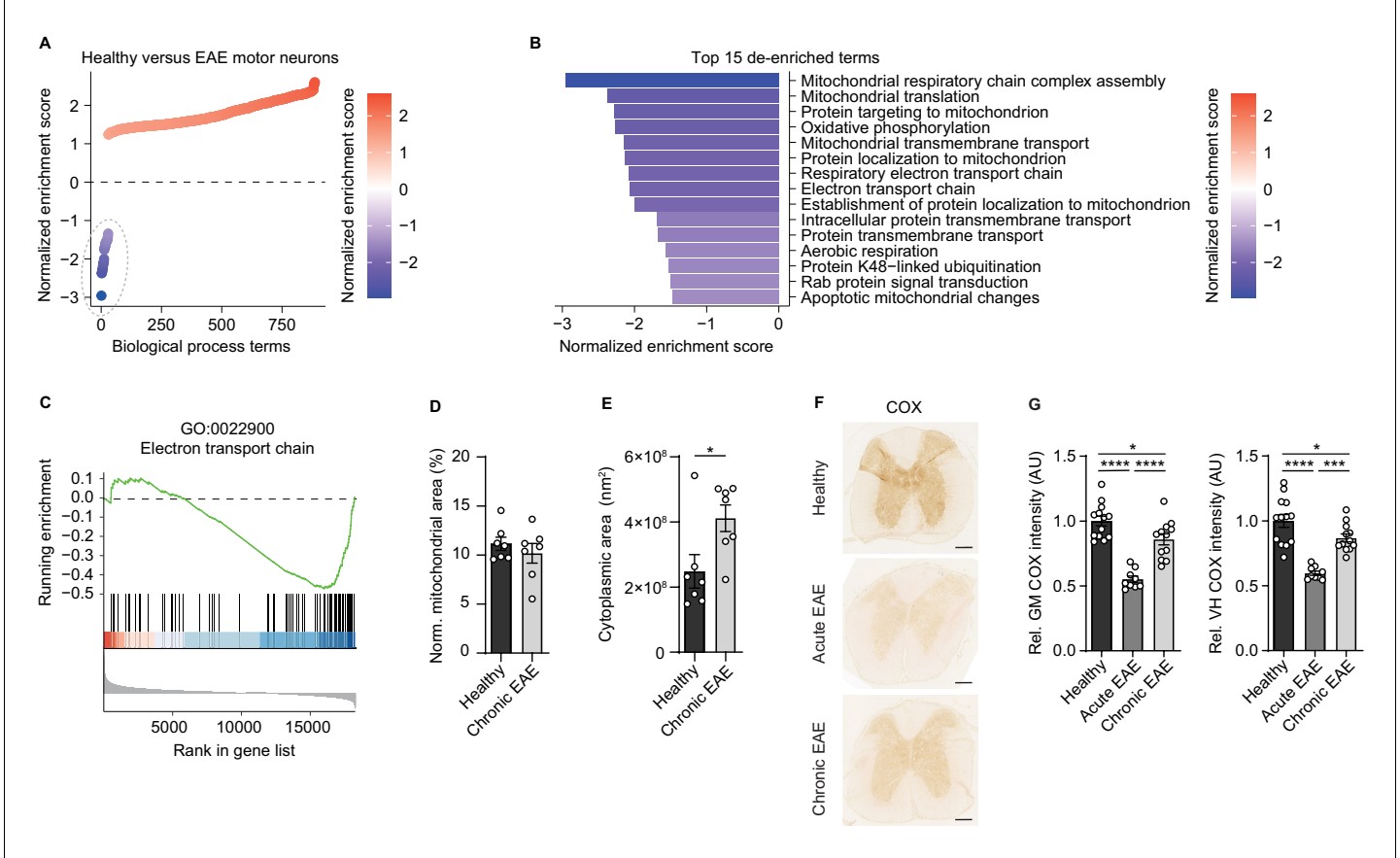

**Figure 1.** Reduced activity of oxidative phosphorylation in spinal cord neurons during experimental autoimmune encephalomyelitis (EAE). (**A**) Gene set enrichment analysis (GSEA) of biological process gene ontology (GO) terms in motor neurons during central nervous system inflammation in the EAE model. Dashed line indicates 29 de-enriched terms. (**B**) Top 15 de-enriched GO terms with normalized enrichment score. (**C**) GSEA plot of the de-enriched term 'GO:0022900 electron transport chain'. (**D**) Transmission electron microscopy (TEM) analysis of motor neuronal mitochondrial content normalized to size of healthy (n = 3) and chronic EAE (n = 3) mice (2–3 cells per mice). Bars show mean values ± s.e.m. (**E**) TEM analysis of motor neuronal size of healthy (n = 3) and chronic EAE (n = 3) mice (2–3 cells per mice). Bars show mean values ± s.e.m. (**F**) Representative images of cytochrome *c* oxidase (COX) histochemistry of cervical spinal cord sections of healthy, acute, and chronic EAE mice. Scale bar: 250 μm. (**G**) Quantification of COX histochemistry of cervical spinal cord gray matter (GM) and ventral horn (VH) of healthy (n = 5), acute (n = 3), and chronic (n = 5) EAE mice (2–3 stainings per mice) normalized to HuC/HuD-positive neurons. Bars show mean values ± s.e.m. Statistical analysis in **D** and **E** was performed by unpaired, two-tailed Student's t-test, and in **G** by one-way ANOVA following Tukey's post-hoc test for multiple comparisons. *p<0.05, ****p<0.0001.

The online version of this article includes the following figure supplement(s) for figure 1:

**Figure supplement 1.** Reduced numbers of neurons and complex IV activity in spinal cord neurons during experimental autoimmune encephalomyelitis (EAE).

## Inactivation of PGC-1α contributes to mitochondrial dysfunction in CNS inflammation

Next we interrogated whether a unifying mechanism can explain the inflammation-induced downregulation of gene transcripts that are involved in ETC with consecutively impaired complex IV activity in neurons at the acute stage of EAE. Notably, we detected that many of the genes that drive the downregulation of the ETC theme are usually induced by the transcriptional coactivator *Ppargc1a* (*Lucas et al., 2014*; *Puigserver et al., 1998*; *Figure 2A*). Consequently, we asked whether *Ppargc1a* could be the sought-after unifying factor that is functionally disturbed. Motor neuronal *Ppargc1a* expression itself was not altered in our previously published dataset comparing acute EAE to controls (*Figure 2B*). After confirming this result by an additional translating ribosome affinity purification of motor neuronal transcripts followed by *Ppargc1a* qPCR (*Figure 2C*) and immunoblot of PGC-1α of the cervical spinal cord of EAE and healthy mice (*Figure 2D, E*), we concluded that

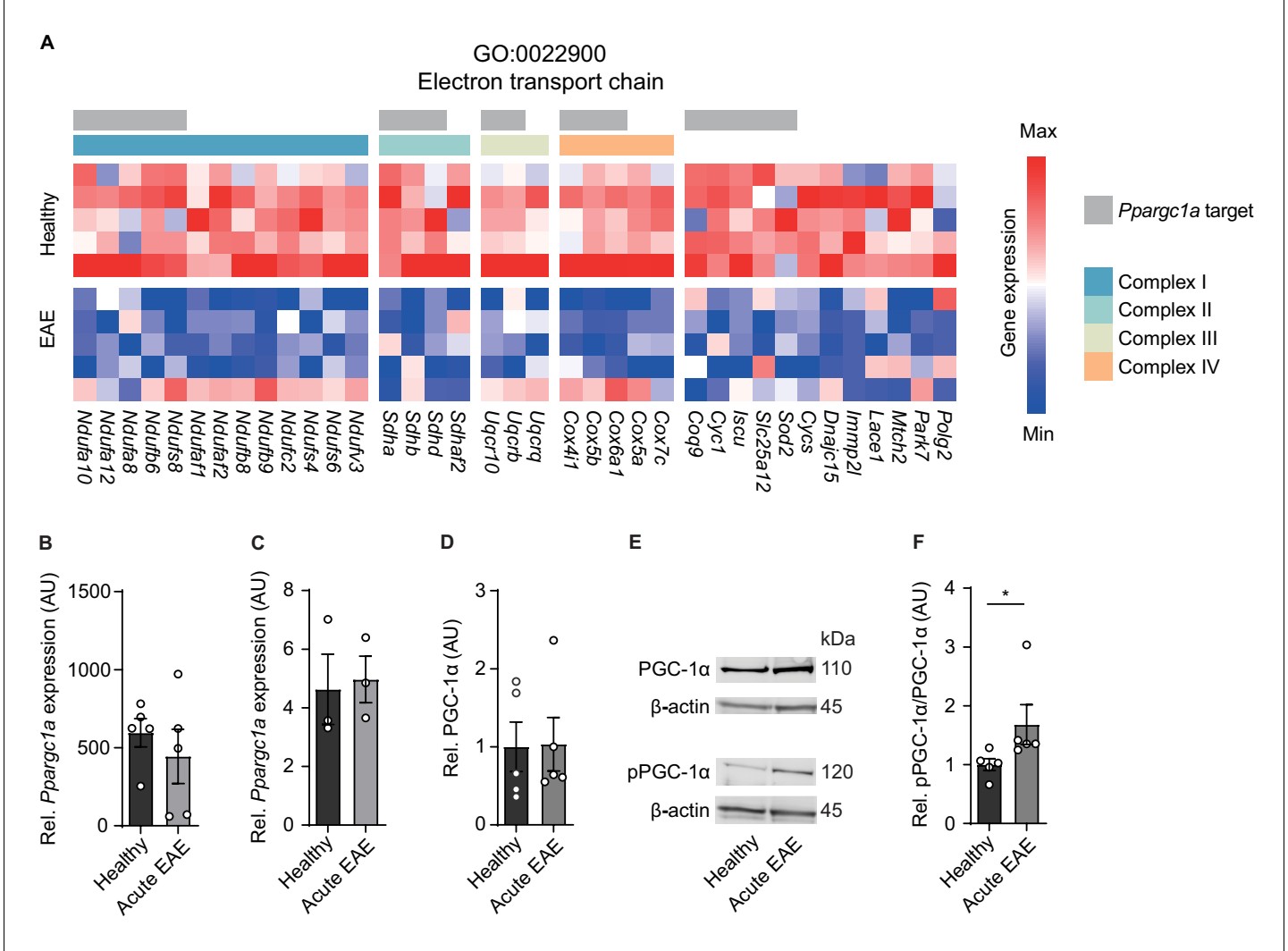

**Figure 2.** Inactivation of proliferator-activated receptor gamma coactivator 1-alpha (PGC-1α) in spinal cord neurons during experimental autoimmune encephalomyelitis (EAE). (**A**) Heatmap of genes driving the de-enrichment of 'GO:0022900 electron transport chain'. Heatmap is annotated for electron transport chain complexes and target genes induced by the transcription factor *Ppargc1a* (*Lucas et al., 2014*). (**B**) Normalized RNA-seq expression of *Ppargc1a* in healthy (n = 5) and acute EAE (n = 5) motor neurons. Bars show mean values ± s.e.m. (**C**) Relative qPCR mRNA expression of *Ppargc1a* to *Tbp* in healthy (n = 3) and acute EAE (n = 3) motor neurons. Bars show mean values ± s.e.m. (**D**) Quantification of PGC-1α protein in cervical spinal cords of healthy (n = 5) and acute (n = 5) EAE mice. Each sample was normalized to its β-actin. Bars show mean values ± s.e.m. (**E**) Representative immunoblots of PGC-1α, phosphorylated PGC-1αS570 (pPGC-1α), and corresponding β-actin of cervical spinal cords of healthy and acute EAE mice. (**F**) Quantification of phosphorylated PGC-1αS570 (pPGC-1α) in relation to PGC-1α total protein in cervical spinal cords of healthy (n = 5) and acute (n = 5) EAE mice. Each sample was normalized to its β-actin. Bars show mean values ± s.e.m. Statistical analysis in **B** and **C** was performed by unpaired, two-tailed Student's t-test, and in **D** and **F** by unpaired, two-tailed Mann–Whitney test; *p<0.05.

The online version of this article includes the following figure supplement(s) for figure 2:

**Figure supplement 1.** Representative immunoblots of phosphorylated PGC-1αS570 (proliferator-activated receptor gamma coactivator 1-alpha [pPGC-1α]) of cervical spinal cords of healthy wild-type mice with and without prior treatment of lambda protein phosphatase for validation of the pPGC-1αS570 antibody.

transcriptional or translational *Ppargc1a*/PGC-1α regulation cannot be the explanation of the observed ETC downregulation.

Besides quantity, PGC-1α can be post-translationally inactivated by phosphorylation at serine 570 (*Li et al., 2007*; *Xiong et al., 2010*), thereby preventing the recruitment of PGC-1α to its regulated promoters, which we tested next. After validation of phosphorylation specificity of the pPGC-1αS570 antibody (*Figure 2—figure supplement 1*), we detected a significant increase in phosphorylated-serine570 PGC-1α in spinal cords of acute EAE mice in comparison to healthy control mice (*Figure 2E, F*). Thus, we concluded that due to the phosphorylation of PGC-1α during acute stage of EAE that is associated with downregulation of its target gene transcripts and compromised mitochondrial function, *Ppargc1a* could be a potential target to rescue mitochondrial function in EAE-affected motor neurons.

## Neuronal *Ppargc1a* induction augments mitochondrial function

Next, we reasoned that elevating neuronal *Ppargc1a* could restore mitochondrial function and serve as a strategy to improve neuronal resilience in CNS inflammation. We utilized transgenic mice with a neuron-specific overexpression of *Ppargc1a* driven under the *Thy1* promoter (Thy1-Ppargc1a) (*Mudò et al., 2012*), which is expressed in mature neurons (*Figure 3—figure supplement 1A*). We could not observe any differences in the health status of these mice. We made sure that elevated transgenic DNA copies of *Ppargc1a* in neurons (*Figure 3—figure supplement 1B*) resulted in elevated mRNA expression of *Ppargc1a* in cortex, hippocampus, and spinal cord as well as in cortical and hippocampal primary neurons of Thy1-Ppargc1a mice in comparison to wild-type mice (*Figure 3—figure supplement 1C–D*). Additionally, we revealed that Thy1-Ppargc1a mice showed an increase of *Ppargc1a*-dependent transcriptional targets in primary neurons (*Figure 3—figure supplement 1E–F*) and *cytochrome c oxidase subunit 4* (*Cox4i1*) and *citrate synthase* (*Cs*) in CNS tissue (*Figure 3—figure supplement 1G–I*). Increased mRNA level led to a corresponding increase in protein amount of PGC-1α in primary neurons (*Figure 3—figure supplement 1J*) and spinal cord, which we confirmed by staining for the transgenically fused FLAG-tag (*Figure 3—figure supplement 1K*). Besides Cox4i1, in primary hippocampal neurons, *Ppargc1a* overexpression induced 11 of the 17 *Ppargc1a* regulated ETC genes, which were downregulated in EAE (*Figure 3A*). Accordingly, neuronal *Ppargc1a* overexpression resulted in increased mitochondrial content in primary neurons (*Figure 3B*) and CNS tissue (*Figure 3C*).

To determine the mitochondrial activity in these mice, we analyzed whether *Ppargc1a*-mediated *Cox4i1* induction resulted in increased complex IV activity. We detected an elevated complex IV activity in the gray matter and ventral horn of Thy1-Ppargc1a mice, which was not due to a higher number of NeuN-positive neurons (*Figure 3D*). Metabolically, primary neurons of Thy1-Ppargc1a mice showed an elevated oxygen consumption rate and maximum respiratory capacity (*Figure 3E*, *Figure 3—source data 1*), which was independent of neuronal numbers (*Figure 3—figure supplement 1L*). In concordance with higher activity of oxidative phosphorylation, primary neurons of Thy1-Ppargc1a mice retained higher tetramethylrhodamin-ethylester (TMRE) fluorescence intensity than wild-type controls, representing a hyperpolarized mitochondrial membrane potential (*Figure 3F*).

## Induction of neuronal *Ppargc1a* improves neuronal calcium buffering capacity

As CNS inflammation results in substantial neuronal calcium influx (*Witte et al., 2019*) leading to neurodegeneration, which could potentially be buffered by mitochondria, we next asked whether increased neuronal mitochondrial activity and membrane potential in Thy1-Ppargc1a neurons could alleviate toxic calcium levels. By analyzing spontaneously active cortical primary neuronal cultures (cPNC) transduced with the virally encoded calcium indicator GCaMP6f (*Figure 4A*), we recorded a significantly faster clearance of intracellular calcium concentrations and decreased amount of mean calcium in spontaneously active Thy1-Ppargc1a primary neurons in comparison to wild-type control neurons, whereas the amplitude and number of calcium transients did not differ (*Figure 4B, C*). The results were confirmed with a chemical calcium indicator (*Figure 4D* and *Video 1*).

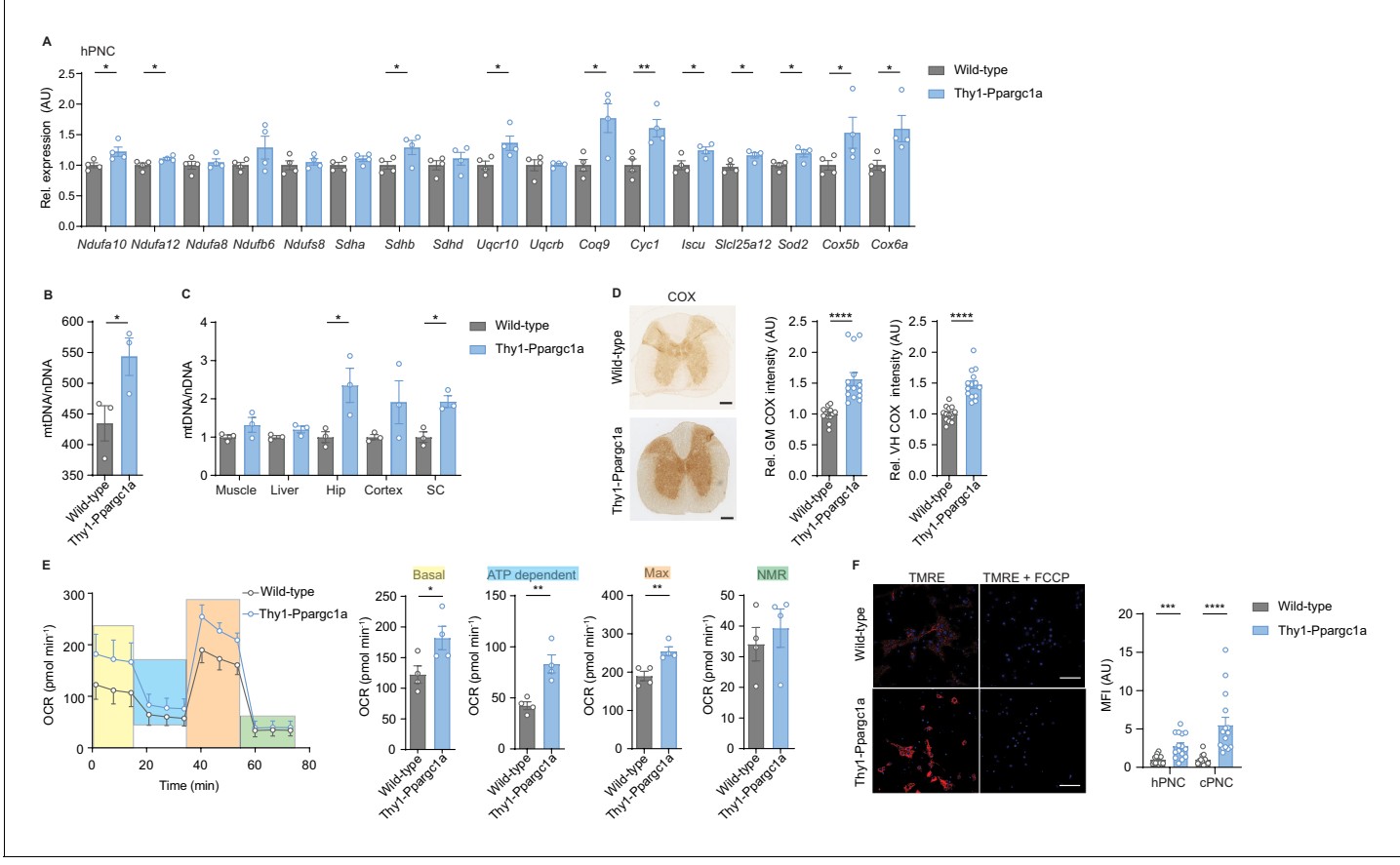

**Figure 3.** Neuronal overexpression of *Ppargc1a* increases neuronal mitochondrial activity. (A) Relative qPCR mRNA expression in hippocampal primary neuronal cultures (hPNC) (DIV14) of wild-type (n = 4) and Thy1-Ppargc1a (n = 4) mice of *Ppargc1a*-regulated electron transport chain genes that were detected to be downregulated during experimental autoimmune encephalomyelitis. Bars show mean values ± s.e.m. (B) Mitochondrial DNA copy numbers (mtDNA) relative to diploid nuclear chromosomal DNA (nDNA) in hPNC (DIV14) of wild-type (n = 3) and Thy1-Ppargc1a (n = 3) mice. Bars show mean values ± s.e.m. (C) mtDNA relative to nDNA in muscle, liver, hippocampus (Hip), cortex, and cervical spinal cord (SC) of wild-type (n = 3) and Thy1-Ppargc1a (n = 3) mice. Bars show mean values ± s.e.m. (D) Representative images and quantification of cytochrome *c* oxidase (COX) histochemistry of cervical spinal cord gray matter (GM) or ventral horn (VH) of wild-type (n = 5) and Thy1-Ppargc1a (n = 6) mice (2–3 stainings per mice) normalized to neuronal nuclei (NeuN)-positive neurons. Bars show mean values ± s.e.m. Scale bar 250 μm. (E) Profile and quantification of oxygen consumption rate in hPNC (DIV14) of wild-type (n = 4) and Thy1-Ppargc1a (n = 4) mice. Yellow: basal respiration (Basal); blue: ATP-dependent respiration (ATP dependent); orange: maximal respiratory capacity (Max); green: non-mitochondrial respiration (NMR). Bars show mean values ± s.e.m. (F) Representative images of hPNC and mean fluorescence intensity quantification of tetramethylrhodamin-ethylester mitochondrial membrane potential assay of hPNC (DIV14) and cortical primary neuronal culture of wild-type (n = 3) and Thy1-Ppargc1a (n = 3) mice (five cells per culture). FCCP (carbonyl cyanide 4-(trifluoromethoxy) phenylhydrazone) was used as ionophore uncoupler of oxidative phosphorylation. Bars show mean values ± s. e.m. Scale bar: 10 μm. Statistical analysis in A was performed by unpaired, one-tailed Student's t-test, and in B–E by unpaired, two-tailed Student's t-test. *p<0.05, **p<0.01, ***p<0.001, ****p<0.0001.

The online version of this article includes the following source data and figure supplement(s) for figure 3:

**Source data 1.** Analysis of oxygen consumption rate in primary neurons of Thy1-Ppargc1a mice.

**Figure supplement 1.** Neuronal *Ppargc1a* and *Ppargc1a*-dependent target genes are increased in Thy1-Ppargc1a mice.

## Elevated neuronal mitochondrial activity ameliorates neurodegeneration in CNS inflammation

As we assumed that inactivation of neuronal *Ppargc1a* during CNS inflammation aggravates mitochondrial dysfunction and spurs neuronal vulnerability, we first tested whether this hypothesis holds true in *Ppargc1a*-deficient neurons. We used *Ppargc1a^flx/flx^ × Eno2^Cre+^* mice to specifically delete *Ppargc1a* in neurons as *Eno2* is widely expressed in neurons (*Figure 5—figure supplement 1A*). Although *Eno2* expression is also present by immune cells (*Heng et al., 2008*), we could exclude *Ppargc1a* expression in immune cells (*Figure 5—figure supplement 1B*, *Heng et al., 2008*),

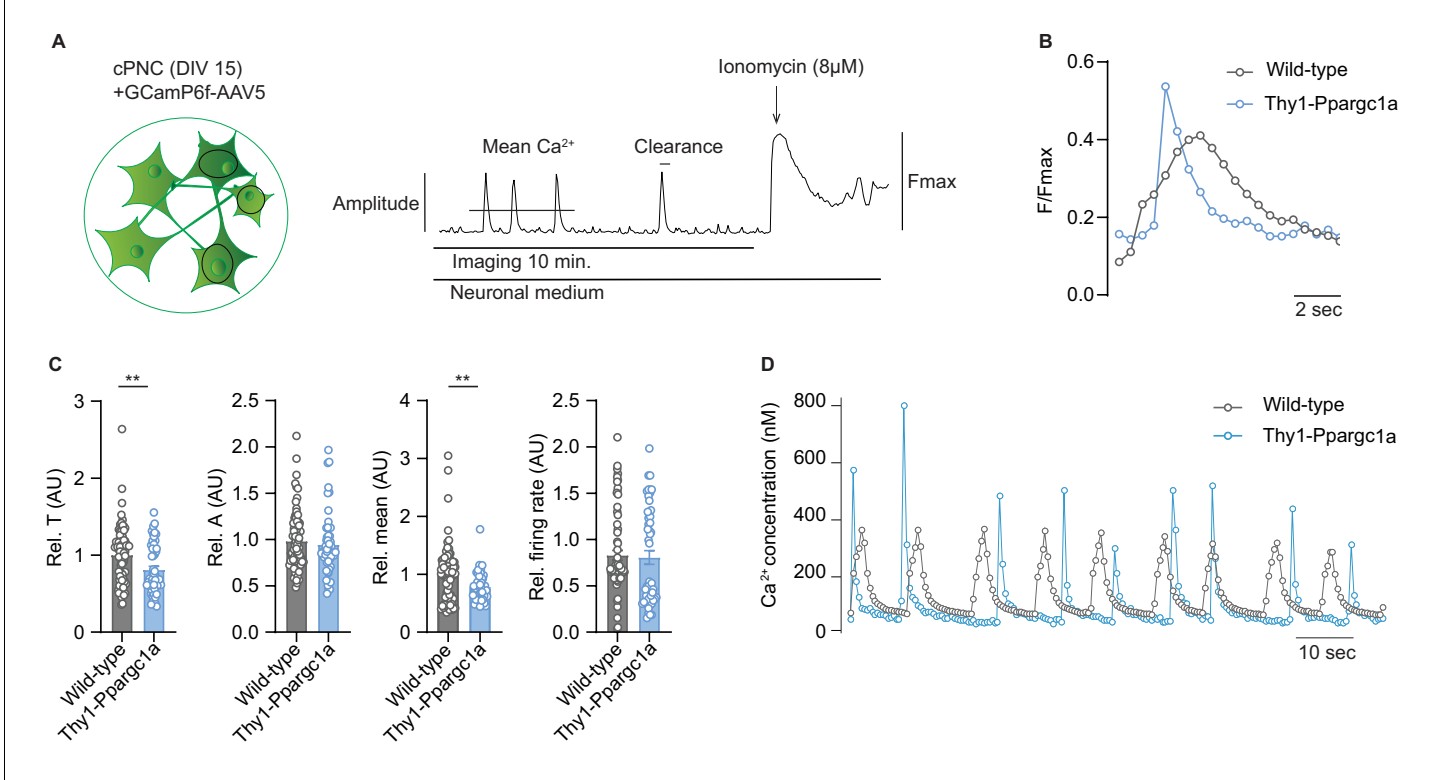

**Figure 4.** Neuronal overexpression of *Ppargc1a* improves neuronal calcium buffering. (**A**) Experimental approach for analysis of $Ca^{2+}$ signaling. GCamp6f fluorescence was recorded for 10 minutes in spontaneously active cortical primary neuronal culture (cPNC) (DIV15) prior to ionomycin application used for signal normalization. (**B**) Representative cytosolic calcium transients of GCaMP6f-transduced spontaneously active cPNC (DIV15) of wild-type and Thy1-Ppargc1a mice normalized to fluorescence of cytosolic calcium-saturated conditions (Fmax). (**C**) Quantification of calcium transient decay constant Tau (T) as an indicator of cytosolic calcium clearance time, calcium transient amplitude (A), mean cytosolic calcium signal intensity and of the number of cytosolic calcium transients presented as firing rate of GCaMP6f-transduced cPNC (DIV15) of wild-type (n = 74 cells from three different mice) and Thy1-Ppargc1a (n = 51 cells from three different mice). 478s of the recorded trace were analysed. Bars show mean values ± s.e.m. (**D**) Representative cytosolic calcium trace of Fluo-4-stained cPNC (DIV14–16) of wild-type and Thy1-Ppargc1a mice. Statistical analysis in **C** was performed by unpaired, two-tailed Student's t-test. \*\*p<0.01.

therefore securing that *Ppargc1a^{flx/flx} × Eno2^{Cre+}* mice will not result in altered immune responses. Neuronal deletion of *Ppargc1a* resulted in decreased *Ppargc1a* mRNA levels (*Figure 5—figure supplement 1C*) and *Ppargc1a*-regulated downstream targets in hippocampus (*Figure 5—figure supplement 1D*), cortex (*Figure 5—figure supplement 1E*), and spinal cord (*Figure 5—figure supplement 1F*). Consequently, neuron-specific *Ppargc1a* knockout mice (*Ppargc1a^{flx/flx} × Eno2^{Cre+}*) showed a more severe EAE disease course (*Figure 5A*, *Figure 5—source data 1*) and by using NeuN-staining an increased neuronal loss in the gray matter and ventral horn in comparison to *Ppargc1a^{flx/flx}* control EAE mice (*Figure 5B*). Since NeuN-staining can be compromised in the chronic phase of EAE, we

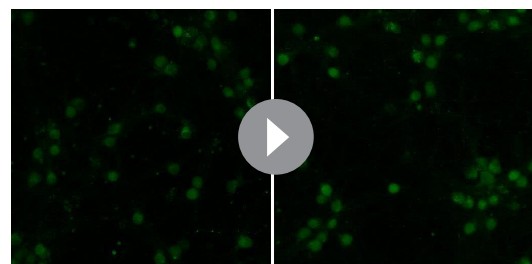

**Video 1.** Representative video of calcium transients of Fluo4 stained cortical primary neuronal culture (DIV15) of Thy1-Ppargc1a (left) and wild-type (right) mice. Enhancing mitochondrial activity in neurons protects against neurodegeneration in a mouse model of multiple sclerosis.
https://elifesciences.org/articles/61798#video1

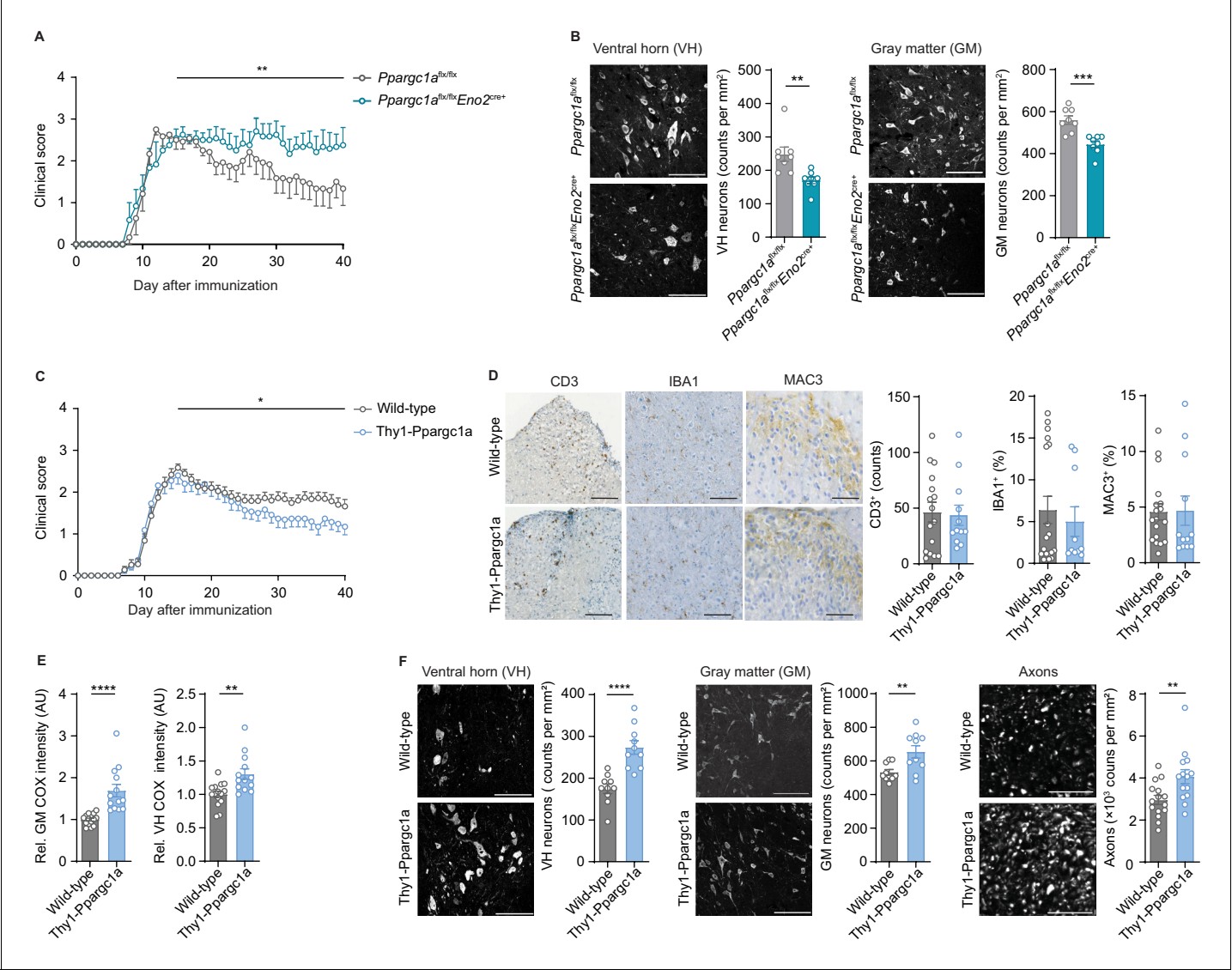

**Figure 5.** Neuronal *Ppargc1a* levels determine neuronal injury and clinical disability in experimental autoimmune encephalomyelitis (EAE). (**A**) Mean clinical scores of *Ppargc1a*$^{flx/flx}$ (n = 6) and *Ppargc1a*$^{flx/fl}$ × *Eno2*$^{Cre+}$ (n = 6) mice undergoing EAE. Curves show mean ± s.e.m. (**B**) Representative images and analysis of immunohistochemical stainings of surviving NeuN-positive neurons in cervical spinal cord ventral horn (VH) or gray matter (GM) of *Ppargc1a*$^{flx/flx}$ (n = 4) and *Ppargc1a*$^{flx/flx}$ × *Eno2*$^{Cre+}$ (n = 4) mice (two areas per mice) at day 40 post-immunization with quantification. Bars show mean values ± s.e.m. Scale bar 100 µm. (**C**) Mean clinical scores of wild-types (n = 31) and Thy1-Ppargc1a (n = 22) mice undergoing EAE. Curves show mean ± s.e.m., all pooled from three independent experiments. (**D**) Histopathological stainings of T cells (CD3), microglia (IBA1), and macrophages (MAC-3) in cervical spinal cord sections of wild-type (n = 6) and Thy1-Ppargc1a (n = 5) mice (2–3 stainings per mice) at day 15 post immunization with quantifications. Bars show mean values ± s.e.m. Scale bar 250 and 100 µm. (**E**) Quantification of cytochrome *c* oxidase (COX) histochemistry of cervical spinal cord GM or VH of wild-type (n = 6) and Thy1-Ppargc1a (n = 4) mice (2–3 stainings per mice) at acute stage of EAE normalized to neuronal nuclei (NeuN)-positive neurons. Bars show mean values ± s.e.m. Scale bar 250 µm. (**F**) Representative images and analysis of immunohistochemical stainings of surviving NeuN-positive neurons in cervical spinal cord VH or GM and surviving neurofilament-positive axons in the dorsal columns of wild-type (n = 5) and Thy1-Ppargc1a (n = 5) (two areas per mice for NeuN, three areas per mice for neurofilament) at day 40 post immunization. Bars show mean values ± s.e.m. Scale bar: 100 µm and 50 µm. Statistical analysis in **A** and **C** was performed by one-tailed Mann–Whitney U test of area under the curve (AUC) starting at peak (day 15) of disease; in **B**, **D**, **E**, and **F** by unpaired, two-tailed Student's t-test. *p<0.05, **p<0.01, ***p<0.001, ****p<0.0001.

The online version of this article includes the following source data and figure supplement(s) for figure 5:

**Source data 1.** Numerical data of clinical scores of *Ppargc1a*$^{flx/flx}$ (flx/flx) (*n* = 6) and *Ppargc1a*$^{flx/flx}$ × *Eno2*$^{Cre+}$ (flx/flx × Eno2Cre) (n = 6) mice undergoing experimental autoimmune encephalomyelitis (EAE).

**Source data 2.** Numerical data of clinical scores of wild-types (WT) (*n* = 31) and Thy1-Ppargc1a (TT) (n = 22) mice undergoing experimental autoimmune encephalomyelitis (EAE).

**Figure supplement 1.** Neuronal *Ppargc1a* and *Ppargc1a*-dependent target genes are decreased in *Ppargc1a*$^{flx/flx}$ × *Eno2*$^{Cre+}$ mice.

corroborated our findings by using the neuronal marker HuC/D as an alternative staining of spinal cord neurons (*Figure 5—figure supplement 1G*). Notably, *Ppargc1a* deletion in neurons did not result in different neuronal numbers in healthy mice by using NeuN staining (*Figure 5—figure supplement 1H*).

As we observed that neuronal *Ppargc1a* overexpression led to an increased oxidative phosphorylation and improved calcium buffering capacities, we next tested its translatability to the preclinical MS model. EAE induction in Thy1-Ppargc1a and wild-type littermate control mice resulted in a comparable maximum score in the acute phase of the disease. However, Thy1-Ppargc1a mice showed a significantly better recovery from clinical disability in comparison to wild-type controls (*Figure 5C*, *Figure 5—source data 2*). Notably, this protection is unlikely to be driven by an impaired immune response as numbers of infiltrating CD3[+] T cells, activated IBA1[+] microglia, and MAC-3[+] macrophages were similar to WT mice (*Figure 5D*). Moreover, neuronal overexpression of *Ppargc1a* resulted in a rescue of complex IV activity during acute EAE in the entire gray matter and ventral horn (*Figure 5E*), leading to a significant reduction of neuronal loss in the gray matter and ventral horn and more intact axons at day 40 post immunization by using NeuN or HuC/HuD as neuronal marker (*Figure 5F*, *Figure 5—figure supplement 1I*). Neuronal *Ppargc1a* overexpression did also not influence neuronal numbers in healthy mice by using NeuN staining (*Figure 5—figure supplement 1J*).

Thus, induction of neuronal mitochondrial activity represents an attractive neuroprotective strategy in CNS inflammation by compensating mitochondrial dysfunction.

## Discussion

Here, we show that neuronal oxidative phosphorylation is compromised during CNS inflammation, which directly contributes to neuronal vulnerability and can be counteracted by induction of neuronal mitochondrial activity.

In an unsupervised survey, we discovered that expression of mitochondrial genes is massively suppressed in motor neurons during CNS inflammation (*Schattling et al., 2019*) and here in particular genes that participate in oxidative phosphorylation. This sparked our interest to explore upstream mechanisms that coordinate this mitochondrial shutdown and whether counter-regulation could rescue neuronal integrity. Our observation is in concordance with other reports that showed alterations of mitochondria in the brain of MS patients with compromised oxidative phosphorylation in demyelinated axons and neurons (*Campbell et al., 2011*; *Mahad et al., 2009*). This could potentially contribute to mitochondrial swelling and dysfunction of axons in EAE (*Nikić et al., 2011*; *Sadeghian et al., 2016*). While we could not detect a decrease in somatic neuronal mitochondrial numbers in the spinal cord during EAE, we discovered a compromised complex IV activity in the spinal cord gray matter. This compromised oxidative phosphorylation during CNS inflammation is likely coordinated by inactivation of PGC-1α, a transcriptional coactivator that acts as a master switch of mitochondrial function (*Mootha et al., 2003*; *Puigserver et al., 1998*). However, in contrast to MS patients, in which a marked decrease in cortical PGC-1α expression has been reported (*Witte et al., 2013*), we could not detect a regulation of *Ppargc1a* in inflamed mouse motor neurons.

By contrast, our data show that CNS inflammation results in a post-translational modification of PGC-1α (*Fernandez-Marcos and Auwerx, 2011*). Increased phosphorylation of serine 570 leads to an inactivation of PGC-1α, which could be mediated by increased AKT activity (*Li et al., 2007*). Three different isoforms of AKT exist. Whereas AKT1 and AKT3 are mainly expressed in neurons, AKT2 is predominantly expressed in astrocytes (*Levenga et al., 2017*). So far, the exact function of AKT in neurons is not clear, but AKT1 has been implicated in late long-term potentiation (*Levenga et al., 2017*) and AKT3 in neuronal growth (*Adams et al., 2016*; *Rivière et al., 2012*). AKT is activated by phosphatidylinositol 3-kinase (PI3K), and the PI3K/AKT signaling pathway is known to reduce apoptosis and promote survival and proliferation (*Brunet et al., 1999*) and thereby counteracts neuronal cell death (*Peltier et al., 2007*). As TNF-α activates the PI3K/AKT pathway (*Gu et al., 2006*; *Li et al., 2017*; *Osawa et al., 2001*), upregulation of AKT could function as an immediate strategy of neurons against cell death during inflammation. As a trade-off, prolonged downregulation of mitochondrial gene transcripts will reduce ATP levels that could limit neuronal integrity during chronic inflammation. Thus, tuning *Ppargc1a* could act as a switch that modulates

oxidative phosphorylation by regulating respective mitochondrial genes under challenging environmental conditions, such as CNS inflammation.

This seems to be particularly relevant in tissues with a high energy demand, for example, muscle, liver, and CNS in which *Ppargc1a* is highly expressed. While astrocytes are mostly relying on glycolysis, neurons mainly generate their energy by oxidative phosphorylation with low glycolytic capacities (*Camandola and Mattson, 2017*). Therefore, mitochondrial damage and inactivation is particularly deleterious for neuronal metabolism. That *Ppargc1a* is important for neuronal health is supported by our observation of an aggravated EAE disease course in neuron-specific *Ppargc1a*-deficient mice. This is in accordance with the global *Ppargc1a* knockout mice that showed neurological symptoms such as myoclonus, dystonia, and limb clasping that was attributed to axonal striatal degeneration (*Lin et al., 2004*), although we were not able to detect this phenotype in our neuron-specific *Ppargc1a* knockout mice.

While a recent report using *Ppargc1a* overexpression under the *Eno2* promoter led to an ameliorated EAE disease course (*Dang et al., 2019*), we believe that this is mainly driven by suppressing the immune response and not by a neuron-intrinsic effect. As *Eno2* is also expressed in immune cells and especially T cell function is influenced by altered mitochondrial activity (*Buck et al., 2016*), in accordance with our notion, the authors reported a reduced immune cell infiltration, less demyelination, a later EAE onset, and an ameliorated EAE disease course (*Dang et al., 2019*), which are typical features for an immunosuppressive phenotype. By contrast, our data show that specific neuronal overexpression of *Ppargc1a* elevated mitochondrial biogenesis and an improved activity of neuronal oxidative phosphorylation, particularly complex IV, which also resulted in a higher overall mitochondrial membrane potential. Notably, these alterations equipped mitochondria with an improved calcium buffering capacity. Besides MS, rise in neuronal intracellular calcium is one of the hallmarks of neurodegenerative diseases (*Mattson, 2007*). Several mechanisms are relevant that lead to intracellular neuronal $Ca^{2+}$ accumulation in EAE (*Siffrin et al., 2015*), among them upregulation of voltage-gated sodium channels, $Na^+$-$Ca^{2+}$ antiporters (*Craner et al., 2004*), $Ca^{2+}$ permeable ion channels (*Friese et al., 2007*; *Schattling et al., 2012*), and nanoscale ruptures of the axonal plasma membrane (*Witte et al., 2019*). Therefore, the improved calcium buffering capacity in Thy1-Ppargc1a neurons could efficiently rescue calcium overload and neuronal demise, which was indicated by an increased neuronal survival during EAE. Calcium uptake by mitochondria holds the promise as a therapeutic strategy for several neurodegenerative diseases (*Lee et al., 2018*; *Parone et al., 2013*). As neuronal mitochondrial deficiency is also associated with other neurodegenerative diseases, for example, Alzheimer's disease (*Pannaccione et al., 2020*), induction of mitochondrial activity could serve as a unifying neuroprotective approach (*Murphy and Hartley, 2018*). Similarly, downstream targets of *Ppargc1a* are repressed in dopaminergic neurons of Parkinson's disease patients (*Zheng et al., 2010*) in which neuronal overexpression of *Ppargc1a* protects dopaminergic neurons in its mouse model (*Mudò et al., 2012*).

As we detected a higher neuronal resilience in Thy1-Ppargc1a mice during CNS inflammation, pharmacological increase of neuronal mitochondria and oxidative phosphorylation could be a promising neuroprotective strategy. Some drugs have already been described to induce *Ppargc1a* in different tissues (*Dumont et al., 2012*; *Hofer et al., 2014*; *Noe et al., 2013*; *Ye et al., 2012*); however, we could not confirm neuronal induction of *Ppargc1a* by treating mice with bezafibrate or using transient receptor potential cation channel subfamily V member 4 (*Trpv4*) knockout mice (data not shown). Another possible drug candidate is sirtuin-1 that activates PGC-1α by deacetylation, and its neuronal overexpression leads to an ameliorated EAE disease course (*Nimmagadda et al., 2013*). Besides extrinsic induction or activation of *Ppargc1a*, it can also be induced by intrinsic mechanisms such as cold exposure or exercise (*Lin et al., 2005*). Exercise ameliorates the EAE disease course (*Klaren et al., 2014*; *Rossi et al., 2009*) and shows several benefits in MS patients (*Motl et al., 2017*); however, the proof of a neuroprotective effect that is mediated by *Ppargc1a* induction is currently outstanding.

Taken together, we provide evidence for a therapeutic potential of inducing mitochondrial activity in inflammation-induced neurodegeneration, supporting further studies that aim at finding drugs to target this pathway in neurons.

# Materials and methods

## Key resources table

| Reagent type (species) or resource | Designation | Source or reference | Identifiers | Additional information |
|---|---|---|---|---|
| Antibody | Alexa Fluor 488 α-chicken (donkey polyclonal) | Jackson | RRID: AB_2340375 | IHC (1:800) |
| Antibody | Alexa Fluor 647 α-rabbit (donkey polyclonal) | Jackson | RRID: AB_2752244 | IHC (1:800) |
| Antibody | α-NeuN (chicken polyclonal) | Millipore | RRID: AB_11205760 | IHC (1:500) |
| Antibody | α-HuC/HuD (mouse monoclonal) | Thermo Scientific | RRID: AB_221448 | IHC (1:500) |
| Antibody | α-Mouse IgG (goat polyclonal) | Jackson | RRID: AB_2338476 | IHC (1:200) |
| Antibody | Cy3 α-mouse (donkey polyclonal) | Jackson | RRID: AB_2340816 | IHC (1:800) |
| Antibody | α-FLAG (Clone M2) (mouse monoclonal) | Sigma | RRID: AB_259529 | IHC (1:200) |
| Antibody | α-SMI31 (mouse monoclonal) | Covance | RRID: AB_10122491 | IHC (1:500) |
| Antibody | α-SMI32 (mouse monoclonal) | Covance | RRID: AB_2564642 | IHC (1:500) |
| Antibody | α-PGC-1α (rabbit polyclonal) | Novus | RRID: AB_1522118 | IHC (1:100) ICC (1:200) WB (1:2000) |
| Antibody | α-MAP2 (chicken polyclonal) | Abcam | RRID: AB_2138153 | ICC (1:2500) |
| Antibody | α-CD3 (clone SP7) (rabbit monoclonal) | Abcam | RRID: AB_443425 | IHC (1:100) |
| Antibody | α-Mac3 (clone M3/84) (rat monoclonal) | BD Biosciences | RRID: AB_394780 | IHC (1:100) |
| Antibody | α-Iba1 (rabbit polyclonal) | Wako | RRID: AB_839504 | IHC (1:100) |
| Antibody | α-Phosphorylated PGC-1αS570 (rabbit polyclonal) | R and D Systems | RRID: AB_10890391 | WB (1:1000) |
| Antibody | α-ß-Actin (rabbit polyclonal) | Cell Signaling Technology | RRID: AB_330288 | WB (1:1000) |
| Antibody | HRP α-rabbit (goat polyclonal) | LI-COR Biosciences | RRID: AB_2721264 | WB (1:15,000) |
| Antibody | α-CD45-APC/Cy7 (clone 30F11) (rat monoclonal) | BioLegend | RRID: AB_312981 | FACS (1:100) |
| Antibody | α-CD4-FITC (clone GK1.5) (rat monoclonal) | BioLegend | RRID: AB_312691 | FACS (1:100) |
| Antibody | α-CD3-BV605 (clone 17A2) (rat monoclonal) | BioLegend | RRID: AB_2562039 | FACS (1:300) |
| Antibody | α-CD8-Pacific Blue (clone 53–6.7) (rat monoclonal) | BioLegend | RRID: AB_493425 | FACS (1:100) |
| Antibody | α-CD19-PE/Cy7 (clone 1D3) (rat monoclonal) | BD | RRID: AB_10894021 | FACS (1:100) |
| Antibody | α-CD19-BV605 (clone 6D5) (rat monoclonal) | BioLegend | RRID: AB_11203538 | FACS (1:100) |

*Continued on next page*

*Continued*

| Reagent type (species) or resource | Designation | Source or reference | Identifiers | Additional information |
|---|---|---|---|---|
| Antibody | α-Ly6G-BV711 (clone 1A8) (rat monoclonal) | BD | RRID: AB_2738520 | FACS (1:100) |
| Antibody | α-F4/80-BV421 (clone T45-2342) (rat monoclonal) | BD | RRID: AB_2734779 | FACS (1:100) |
| Antibody | α-NK1.1- PE/Cy7 (clone PK136) (rat monoclonal) | BioLegend | RRID: AB_389364 | FACS (1:300) |
| Antibody | α-CD11b-FITC (clone M1/70) (rat monoclonal) | BioLegend | RRID: AB_312789 | FACS (1:300) |
| Antibody | α-Fc-Block (true stain anti-mouse CD16/32, clone 93) (rat monoclonal) | BioLegend | RRID: AB_1574975 | FACS (1:1000) |
| Chemical compound, drug | *Mycobacterium tuberculosis* | BD Difco | BD 231141 | |
| Chemical compound, drug | Freund's adjuvant | Difco Laboratories | Cat. number: 263910 | |
| Chemical compound, drug | Dynabeads MyOne Streptavidin T1 | Invitrogen | Cat. number: 65601 | |
| Chemical compound, drug | Cytochrome *c* from bovine heart | Sigma-Aldrich | CAS number: 9007-43-6 | |
| Chemical compound, drug | Bis(2-amino-phenoxy) ethane tetraacetic acid | Thermo Scientific | CAS number: 126150-97-8 | |
| Chemical compound, drug | 3,3-Diaminobenzidine (DAB),$\geq$98% | Sigma-Aldrich | Cat. number: D8001 | |
| Chemical compound, drug | Fluo-4, AM | Invitrogen | Cat. number: F14201 | |
| Commercial assay or kit | Bicinchoninic acid (BCA) kit for protein determination | Sigma-Aldrich | Cat. number: BCA1 | |
| Commercial assay or kit | Seahorse XF cell mito stress test kit | Agilent | Part number: 103015-100 | |
| Commercial assay or kit | Calcein-AM | Sigma-Aldrich | CAS number: 14850434–1 | |
| Commercial assay or kit | ultraView Universal DAB detection kit | Ventana | Cat. number: 760-500 | |
| Commercial assay or kit | TMRE-mitochondrial membrane potential assay kit | Abcam | Ab113852 | |
| Genetic reagent (*Mus musculus*) | C57BL/6J wild-type mice | Jackson Laboratory | Stock #: 000664 RRID: MGI:2159769 | |
| Genetic reagent (*M. musculus*) | Thy1-Flag-Ppargc1a | PMID: 21984601 | | Dr. Dan Lindholm (Minerva Medical Research Institute) |
| Genetic reagent (*M. musculus*) | Eno2$^{Cre+}$ | Jackson Laboratory | Stock #: 006663 RRID: MGI:2177175 | |
| Genetic reagent (*M. musculus*) | Ppargc1a$^{flx/flx}$ | Jackson Laboratory | Stock #: 009666 RRID: MGI:5576884 | |
| Genetic reagent (*M. musculus*) | Chat-L10a-eGFP | Jackson Laboratory | Stock #: 030250 RRID: MGI:5496680 | |
| Other | Bolt 4–12% Bis-Tris Plus Gel | Invitrogen | Cat. number: NW04120BOX | |
| Peptide, recombinant protein | MOG$_{35-55}$ peptide | Peptides and elephants | Order number: EP02030_1 | |
| Sequence-based reagent | *Actb* | Thermo Scientific | Mm_00607939_s1 | |

*Continued on next page*

*Continued*

| Reagent type (species) or resource | Designation | Source or reference | Identifiers | Additional information |
|---|---|---|---|---|
| Sequence-based reagent | *Alas* | Thermo Scientific | Mm01235914_m1 | |
| Sequence-based reagent | *Coq9* | Thermo Scientific | Mm00804236_m1 | |
| Sequence-based reagent | *Cox2l* | Thermo Scientific | Mm03294838_g1 | |
| Sequence-based reagent | *Cox4il* | Thermo Scientific | Mm01250094_m1 | |
| Sequence-based reagent | *Cox5a* | Thermo Scientific | Mm00432638_m1 | |
| Sequence-based reagent | *Cox5b* | Thermo Scientific | Mm01229713_g1 | |
| Sequence-based reagent | *Cox6a1* | Thermo Scientific | Mm01612194_m1 | |
| Sequence-based reagent | *Cs* | Thermo Scientific | Mm00466043_m1 | |
| Sequence-based reagent | *Cyc1* | Thermo Scientific | Mm00470540_m1 | |
| Sequence-based reagent | *Iscu* | Thermo Scientific | Mm02342800_g1 | |
| Sequence-based reagent | *Ndufa10* | Thermo Scientific | Mm00600325_m1 | |
| Sequence-based reagent | *Ndufa12* | Thermo Scientific | Mm01240336_m1 | |
| Sequence-based reagent | *Ndufa8* | Thermo Scientific | Mm00503351_m1 | |
| Sequence-based reagent | *Ndufb6* | Thermo Scientific | Mm07294890_m1 | |
| Sequence-based reagent | *Ndufs8* | Thermo Scientific | Mm00523063_m1 | |
| Sequence-based reagent | *Nrf1* | Thermo Scientific | Mm01135606_m1 | |
| Sequence-based reagent | *Ppargc1a* | Thermo Scientific | Mm00464452_m1 | |
| Sequence-based reagent | *Sdha* | Thermo Scientific | Mm01352366_m1 | |
| Sequence-based reagent | *Sdhb* | Thermo Scientific | Mm00458272_m1 | |
| Sequence-based reagent | *Sdhd* | Thermo Scientific | Mm00546511_m1 | |
| Sequence-based reagent | *Slc25a12* | Thermo Scientific | Mm00552467_m1 | |
| Sequence-based reagent | *Sod2* | Thermo Scientific | Mm01313000_m1 | |
| Sequence-based reagent | *Tbp* | Thermo Scientific | Mm01277042_m1 | |
| Sequence-based reagent | *Tfam* | Thermo Scientific | Mm00447485_m1 | |
| Sequence-based reagent | *Uqcr10* | Thermo Scientific | Mm01186961_m1 | |
| Sequence-based reagent | *Uqcrb* | Thermo Scientific | Mm01615741_gH | |

*Continued on next page*

*Continued*

| Reagent type (species) or resource | Designation | Source or reference | Identifiers | Additional information |
|---|---|---|---|---|
| Sequence-based reagent | *Ppargc1a* | Thermo Scientific | Mm00164544_cn, FAM | |
| Sequence-based reagent | *Tfrc* | Thermo Scientific | TaqMan Copy Number reference Assay for mouse, VIC | |
| Transfected construct (synthetic) | pAAV.Syn.GCaM P6f.WPRE.SV40 | Abcam | RRID: Addgene_100837 | Cytosolic calcium indicator |

## Mice

All mice (C57BL/6J wild type [The Jackson Laboratory, Bar Harbor, USA], Thy1-Flag-Ppargc1a [Thy1-Ppargc1a] mice on a C57BL/6J genetic background provided by D. Lindholm [*Mudò et al., 2012*], *Eno2*$^{Cre+}$ mice [The Jackson Laboratory], *Ppargc1a*$^{flx/flx}$ mice [The Jackson Laboratory], and *Chat*-L10a-eGFP mice [*Heiman et al., 2008*]) were kept under specific pathogen-free conditions in the central animal facility of the University Medical Center Hamburg-Eppendorf, Hamburg, Germany. Neuron-specific knockout mice were generated by crossing *Ppargc1a*$^{flx/flx}$ with *Eno2*$^{Cre+}$ mice. Mice were grouped housed in a facility with 55–65% humidity at $24 \pm 2°C$ with a 12 hr light/dark cycle and had free access to food and water. Sex- and age-matched adult animals (10–20 weeks of age) were used in all experiments. Wild-type mice were allocated to EAE or healthy group by cages. All procedures were carried out in accordance with the ARRIVE guidelines (*Kilkenny et al., 2010*).

## EAE induction

Mice were anesthetized with isoflurane 1–2% v/v oxygen and immunized subcutaneously with 200 µg myelin oligodendrocyte glycoprotein 35–55 (MOG$_{35-55}$) peptide (peptides and elephants) in complete Freund's adjuvant (BD) containing 4 mg/ml *Mycobacterium tuberculosis* (BD, Franklin Lakes, USA). A 200 ng pertussis toxin (Merck, Darmstadt, Germany) was injected intravenously on the day of immunization and 48 hr later. Animals were scored daily for clinical signs by the following system: 0, no clinical deficits; 1, tail weakness; 2, hind limb paresis; 3, partial hind limb paralysis; 3.5, full hind limb paralysis; 4, full hind limb paralysis and forelimb paresis; and 5, premorbid or dead. Animals reaching a clinical score ≥4 were sacrificed according to the regulations of the Animal Welfare Act. The experimenters were blinded to the genotype until the end of the experiment, including data analysis. Sex- and age-matched adult animals (8–12 weeks of age) were used in all experiments. For Thy1-Ppargc1a mice and wild-type controls, three independent EAE experiments were conducted, and the data were pooled for final analysis. For *Ppargc1a*$^{flx/flx}$ × *Eno2*$^{Cre+}$ EAE and *Ppargc1a*$^{flx/flx}$ controls, one EAE was conducted. For analysis of the disease course and weight, we only included animals that received a score ≥1 until day 15 and survived until the end of the experiment. Animals were either analyzed at acute stage of EAE (day 12 to day 16 after immunization) or chronic stage (day 30 to day 40 after immunization).

## Gene set enrichment analysis

Gene signature of motor neurons during CNS inflammation was generated by ranking all expressed genes by the DESeq2-derived t statistics based on the comparison of healthy versus EAE motor neurons (*Schattling et al., 2019*). Enrichment analysis was performed using the function 'gseGO' of the R package clusterProfiler (*Yu et al., 2012*) on biological process GO terms with at least 50 members. Gene sets with a Benjamini–Hochberg adjusted p value <0.05 were considered significant and ordered by their normalized enrichment score, with positive values indicating enrichment and negative values indicating de-enrichment. Core enrichment genes driving the de-enrichment of the term 'GO:0022900 electron transport chain' were extracted from clusterProfiler results and plotted as heatmap of gene expression values after variance stabilizing transformation. Plotting was performed with the R packages ggplot2 (https://ggplot2.tidyverse.org; *Wickham, 2016*), clusterProfiler and tidyheatmaps (*Engler, 2020*).

### *Ppargc1a* target identification

For the identification of *Ppargc1a*-induced downstream targets, a published microarray dataset of *Ppargc1a*-overexpressing SH-SY5Y cells was used (*Lucas et al., 2014*). The expression matrix was downloaded via GEOquery (GSE100341) and analyzed using limma. Genes with a false discovery rate (FDR)-adjusted p value <0.05 and a log2 fold change >1 in *Ppargc1a* overexpression were considered positively regulated *Ppargc1a* targets.

## Mouse tissue preparation and histopathology of mice

Mice were anesthetized intraperitoneally with 100 μl solution (10 mg/ml esketamine hydrochloride [Pfizer, New York City, USA], 1.6 mg/ml xylazine hydrochloride [Bayer, Leverkusen, Germany] dissolved in water) per 10 g of body weight. For histopathology and immunohistochemistry, mice were perfused with 4% paraformaldehyde (PFA), cervical spinal cord tissue was dissected, fixed for 30 min with 4% PFA, and then transferred to 30% sucrose in phosphate-buffered saline (PBS) at 4°C. Cervical spinal cord was used as it was shown that clinical scores correlate better with cervical than lumbar spinal cord lesions in EAE (*Fournier et al., 2017*). We transversely cut midcervical spinal cord sections at 12 μm with a freezing microtome (Leica Jung CM3000) and stored them in cryoprotective medium (Jung) at –80°C. For transmission electron microscopy, mice were perfused with 4% PFA, 1% glutaraldehyde (GA) in 0.1 M phosphate buffer, and spinal cords were dissected and fixed with 2% PFA, 2.5% GA in 0.1 M phosphate buffer. Antibodies directed against CD3 (rabbit IgG, Abcam, Cambridge, UK; RRID: AB_443425), Mac-3 (rat IgG, BD Biosciences; RRID: AB_394780), and Iba1 (rabbit, Wako, Richmond, USA; RRID: AB_839504) were visualized by avidin-biotin technique with 3,3-diaminobenzidine (DAB, Sigma, St.Louis, USA) according to standard procedures of the UKE Mouse Pathology Facility using the ultraView Universal DAB Detection Kit (Ventana, Oro Valley, USA). For complex IV/cytochrome *c* oxidase (COX) histochemistry, sections were incubated for 60 min at 37°C with COX reaction media (diaminobenzidine tetrahydrochloride, cytochrome *c* and bovine catalase [all from Sigma] in 0.2 M phosphate buffer) and embedded with Aquatex (Merck). Images of tissue sections at ×200 magnification were scanned using Zeiss MIRAX MIDI Slide Scanner (Carl Zeiss, MicroImaging GmbH, Germany). CD3$^+$, IBA1$^+$, and MAC-3$^+$ infiltrating cells were analyzed with ImageJ software (NIH, Bethesda, USA), with the same settings across all experimental groups. CD3$^+$ cells were counted in entire spinal cord sections. For IBA1$^+$ and MAC3$^+$ quantification, we calculated the percentage of the positively stained area in the spinal cord. For quantification of complex IV activity, background was subtracted and mean gray value intensities of COX reactivity in complete gray matter or the ventral horn were measured by ImageJ. COX reactivity was normalized to corresponding neuronal count.

## Immunohistochemistry (IHC)

Sections were incubated in blocking solution (10% normal donkey serum in PBS) containing 0.05% Triton X-100 at room temperature for 45 min, incubated in 0.05% Triton X-100 containing a Fab Fragment anti-mouse IgG (goat, 1:200; Jackson; RRID: AB_2338476) for 60 min in case of staining anti-mouse primary antibodies to prevent unspecific binding, and subsequently stained them overnight at 4°C with antibodies against the following structures: phosphorylated neurofilaments (SMI 31, mouse, 1:500; Covance, Princeton, USA; RRID: AB_10122491), non-phosphorylated neurofilaments (SMI 32, mouse, 1:500; Covance; RRID: AB_2564642), neuronal nuclei (NeuN, chicken 1:500; Millipore, Burlington, USA; RRID: AB_11205760), neuronal protein (HuC/D, mouse 1:500; Thermo Fisher Scientific, Waltham, USA; RRID: AB_221448), PGC-1α (PGC-1α, rabbit, 1:100; Novus Biologicals, Littleton, USA; RRID: AB_1522118), and FLAG sequence (FLAG, mouse, 1:200; Sigma; RRID: AB_259529). As secondary antibodies we used Alexa Fluor 488-coupled donkey antibodies recognizing chicken IgG (1:800, Jackson; RRID: AB_2340375), Cy3-coupled donkey antibodies recognizing mouse IgG (1:800, Jackson; RRID: AB_2340816), and Alexa Fluor 647-coupled donkey antibodies recognizing rabbit IgG (1:800, Abcam; RRID: AB_2752244). We analyzed the sections with a Zeiss LSM 700 confocal microscope. For quantification of neurons and axons, tile scans of each animal were taken. Numbers of neurons were manually counted in two defined areas (250 μm × 250 μm) from the central gray matter or ventral horn. For absolute numbers of neurons in the ventral horn, all neurons in both ventral horns per animals were counted. Axons were quantified in three

defined areas (90 μm × 90 μm) with ImageJ software using fixed threshold intensity across experimental groups for each type of tissue examined.

## Immunocytochemistry (ICC)

Cortical neurons (DIV14) were fixed with 4% PFA for 30 min at room temperature, permeabilized with 0.05% Triton and blocked with 10% normal donkey serum in PBS. Cells were stained overnight at 4°C with antibodies directed against PGC-1α (PGC-1α, rabbit, 1:200; Novus Biologicals; RRID: AB_1522118) and microtubule-associated protein 2 (MAP2, chicken, 1:2500; Abcam; RRID: AB_2138153). As secondary antibodies we used Alexa Fluor 488-coupled donkey antibodies recognizing chicken IgG (1:800, Jackson; RRID: AB_2340375), Cy3-coupled donkey antibodies recognizing mouse IgG (1:800, Jackson; RRID: AB_2340816), and Alexa Fluor 647-coupled donkey antibodies recognizing rabbit IgG (1:800, Abcam; RRID: AB_2752244). Images were taken with a Zeiss LSM 700 confocal microscope.

## Transmission electron microscopy

After dissection and fixation, spinal cords were cut in 1 mm transverse sections and post-fixed in 1% osmium tetroxide ($OsO_4$) for 1 hr. Hereafter, samples were dehydrated in serial dilutions of alcohol and embedded with Epon (LX-112 Resin, Ladd Research, USA). Next, ultrathin (50-85 nm) sections were collected on 200 mesh copper grids and counterstained with uranyl acetate and lead citrate. Images of individual motor neurons were taken at ×5000 magnification using a JEM 1010 transmission electron microscope (JEM 1010, Jeol USA). From each motor neuron, we quantified the surface area of neuronal cytoplasm and all cytoplasmic mitochondria using ImageJ (NIH), which were used to calculate mitochondrial density per neuron.

## Primary neuronal culture

Primary hippocampal and cortical cultures were prepared from E16.5 embryos. Hippocampus and cortex were harvested, cut into smaller pieces, and incubated in 0.05% Trypsin-EDTA (Gibco) for 6 min at 37°C. Trypsination was stopped by DMEM-F12 containing 10% fetal calf serum (FCS). Afterwards, tissue was dissociated in Hank's balanced salt solution (HBSS) and centrifuged for 2 min at 500 × g. The pellet was resuspended in primary growth medium (PGM), and cells were plated at $5 \times 10^4$ per $cm^2$ (hippocampal neurons) or $7.5 \times 10^4$ per $cm^2$ (cortical neurons) on poly-d-lysine-coated cell culture plates. We maintained cells in PNGM (Primary Neuron Growth Medium BulletKit, Lonza) or neurobasal plus medium (supplemented with B27, penicillin, streptomycin and L-glutamine; Gibco) at 37°C, 5% $CO_2$, and a relative humidity of 98%. To inhibit glial proliferation, we added cytarabine (Sigma, 5 μM) or floxuridine/uridine (Sigma, 10 μM) at 3–4 days in vitro (DIV) at 5 μM and maintained cultures for 14–21 days in vitro (DIV14 or DIV21).

## Calcium imaging

Cortical primary neurons were transduced with genetically encoded cytosolic calcium indicator GCaMP6f (Addgene 100837) via adeno-associated virus (AAV7) at day 7 in vitro. Calcium imaging was performed from cPNC from E16.5 embryos (DIV15). Alternatively, chemical calcium indicator Fluo4-AM (Thermo Fisher, F14201) was used according to the manufacturer's instructions. Briefly, cultures were incubated in optic buffer ($ddH_2O$ supplemented with 10 mM glucose, 140 mM NaCl, 1 mM $MgCl_2$, 5 mM KCl, 20 mM HEPES, 2 mM $CaCl_2$, pH 7.3 adjusted with NaOH) with 4 μM Fluo4-AM for 30 min at room temperature, washed and imaged in optic buffer. Live cell imaging was performed at day 15 in vitro with a Zeiss LSM 700 confocal microscope, 20× magnification (objective) and temporal resolution of 478 ms. Transduced cultures were placed to incubation chamber (33°C, 5% $CO_2$) 20 min prior to imaging and then recorded. Ionomycin (8 μM) was added at the end of each experiment for signal normalization and cultures were imaged for another 15 min to allow development of maximum signal intensity. Additional normalization with optic buffer without $Ca^{2+}$ and with 50 μM BAPTA-AM (Thermo Fisher) was done for the experiments using Fluo4-AM as calcium indicator to determine absolute calcium concentration (*Yasuda et al., 2004*). The 478 s of spontaneous $Ca^{2+}$ transients before application of ionomycin were used for analysis. Time-series images were generated for further analysis. Manual image segmentation and generation of traces of the intensity from the single neuronal soma was performed in ImageJ. Normalization, calcium spikes,

base line detection, and analysis of the calcium clearance rate were performed with the custom-made script written on Python 3.6 (https://github.com/scriptcalcium/PGC1alpha; *Rosenkranz, 2021*; copy archived at swh:1:rev:af3c206a43f8b6e9fdbd7707c9b4601287d5968b) (*Shaposhnykov, 2020*). Clearance time (T), amplitude (A), mean signal intensity (Mean), and numbers of calcium transients = firing rate (N) were defined as follows:

$$N_i = n \; if \; (F(n) \geq F(a), where \; a \in [n - \delta \ldots n + \delta]) \; and \left( F(n) > \frac{1 + \gamma}{2\theta + 1} \sum_{j=n-\theta}^{n+\theta} F(j) \right),$$

Firing rate ($N_i$): N: calcium transient count; F: normalized signal intensity; i and n: indexes for defined frame (n) and cell (i); parameters $\delta$, $\theta$, and $\gamma$ were set as 3, 7, and 0.25 for all analyzed cultures by manual assessment of effective pick detection >80%

$$T_i = \frac{1}{k_i} \sum_{j=1}^{k_i} T_j$$

Clearance time ($T_i$): clearance time for the ith cells with detected k transients, from maximum to half of the amplitude

$$A_i = \frac{1}{k_i} \sum_{j=1}^{k_i} Fp_{ij}$$

Amplitude ($A_i$): amplitude for the ith cell with detected k transients; $Fp_i$ – amplitude of individual transients of ith cell

$$Mean_{[a,b]} = \frac{1}{b - a} \sum_{j=a}^{b} F(j)$$

Mean$_{[a,b]}$: mean of recorded signal from frame a till frame b, F(j) – signal extracted from jth frame; parameters a and b were determined for each experiment individually for each experiment with a constant analyzed interval; last frame was recorded before ionomycin addition.

## Live cell metabolic assay

Hippocampal neurons were seeded in a poly-d-lysine-coated XF 96-well cell culture microplate (Seahorse Bioscience, Copenhagen, Denmark) in triplicate at $5 \times 10^5$ cells per well in 1 ml neuronal growth medium and then incubated at 37°C in 5% $CO_2$. At DIV14, media from neurons was removed, replaced by 180 μl of assay media (Assay Media from Bioscience with 25 mM glucose, 1 mM sodium pyruvate, and 2 mM L-glutamine; pH 7.4), and incubated in a $CO_2$-free incubator at 37°C for 1 hr. Compounds (2 μM oligomycin, 1 μM FCCP, 0.5 μM rotenone/antimycin A, all in assay media) were added into the appropriate ports of a hydrated sensor cartridge. For baseline measurement, control ports were left without compound addition. Cell plate and cartridge were then placed into the XFe96 Analyzer and results analyzed by WAVE Software.

## Cell viability assay (calcein)

Hippocampal neurons were seeded in a poly-d-lysine-coated XF 96-well cell culture microplate (Seahorse Bioscience, Copenhagen, Denmark) in quadruplicate at $5 \times 10^5$ cells per well in 1 ml neuronal growth medium and then incubated at 37°C in 5% $CO_2$. At DIV14, calcein-AM (Sigma) was added to the cultures (2 μM), incubated at 37°C in 5% $CO_2$ for 30 min, washed in optic buffer (ddH$_2$O supplemented with 10 mM glucose, 140 mM NaCl, 1 mM MgCl$_2$, 5 mM KCl, 20 mM HEPES, 2 mM CaCl$_2$, pH 7.3 adjusted with NaOH) at room temperature, and fluorescence intensity was measured according to the manufacturer's instructions.

## Mitochondrial membrane potential assay

Primary neurons were seeded on μ-dishes (ibidi) and TMRE assay (Abcam) was performed on DIV14 according to the manufacturer's protocol. Cells were treated with 10 nM TMRE for 20 min in PGM, washed twice with HBSS w/o phenol red (Gibco), and then equilibrated for 10 min in HBSS w/o

phenol red and live imaged at 555 nm with a Zeiss LSM 700 confocal microscope. For internal control, 20 µM FCCP was added 10 min prior to TMRE. Images were taken with a Zeiss LSM 700 confocal microscope. Total fluorescence of five cells per biological replicate was analyzed.

## Translating ribosome affinity purification

Chat-L10a-eGFP mice were anesthetized with ketamin/xylazin and perfused with 10 ml ice-cold dissection buffer (1 × HBSS, 2.5 mM HEPES-KOH pH 7.4, 35 mM glucose, 4 mM NaHCO$_3$) over 1 min. Cervical spinal cords were dissected in ice-cold dissection buffer containing 100 µg/ml cycloheximide, and three spinal cords were pooled for homogenization in lysis buffer (20 mM HEPES-KOH pH 7.4, 150 mM KCl, 5 mM MgCl$_2$, 0.5 mM dithiothreitol, 100 µg/ml cycloheximide, 40 U/ml Rnasin, 20 U/ml Superasin) using a glass homogenizer. Homogenates were centrifuged at 2000 × g for 10 min at 4°C to remove large cell debris, supernatant was transferred to a new tube, and NP-40 detergent solution (Thermo Fisher Scientific) and 1,2-diheptanoyl-sn-glycero-3-phosphocholine (Avanti Polar Lipids) were added to final concentrations of 1% and 30 mM, respectively. After 5 min incubation on ice, lysates were centrifuged at 20,000 × g for 10 min at 4°C. Ten percent of the supernatant was saved as input control, the remaining 90% were incubated with monoclonal GFP antibody (Htz-GFP19C8 and Htz-GFP19F7; Memorial Sloan Kettering Cancer Center Monoclonal Antibody Core Facility)-coated magnetic beads (Streptavidin MyOne T1 Dynabeads, Invitrogen pre-coupled to biotinylated Protein L, Pierce) with end-over-end rotation overnight at 4°C. Beads were subsequently collected on a magnetic rack, washed four times with high-salt wash buffer (20 mM HEPES, 350 mM KCl, 5 mM MgCl$_2$, 0.5 mM, 1% NP-40, 0.5 mM dithiothreitol, 100 µg/ml cycloheximide), and immediately subjected to Trizol/chloroform-based RNA extraction (Invitrogen). RNA was precipitated with sodium acetate and Glycoblue (Ambion, Thermo Fisher Scientific) in isopropanol overnight at –80°C, washed twice with 70% ethanol, resuspended in water, and further purified using the RNeasy Micro Kit (Qiagen, Venlo, The Netherlands) with on-column DNaseI digestion. For higher RNA yields, all steps were carried out in non-stick microfuge tubes (Ambion).

## Fluorescence-activated cell sorting (FACS)

Mice were anesthetized and killed by inhalation of CO$_2$, and spleens were excised with sterile instruments and collected in ice-cold PBS. Pooled single-cell suspensions from spleen were prepared by homogenization through a 40 µm cell strainer. Cells were pelleted by centrifugation (300 × g, 10 min, 4°C), and lysis of splenic erythrocytes was initiated by RBC lysis buffer (0.15 M NH$_4$Cl, 10 mM KHCO$_3$, 0.1 mM Na$_2$EDTA, pH = 7.4) for 2.5 min at 4°C and stopped with MACS buffer (PBS, 0.5% bovine serum albumin (BSA), 2 mM EDTA). For fluorescence-activated cell sorting, cells were washed with PBS and stained with Fixable Viability Stain 700 (BD Biosciences) in PBS for 20 min at 4°C in the dark to exclude dead cells. Cell suspension was washed in PBS. Cells were stained with CD45-APC/Cy7 (clone 30F11, BioLegend; RRID: AB_312981), CD3-BV605 (clone 17A2, BioLegend; RRID: AB_2562039), CD4-FITC (clone GK 1.5, BioLegend; RRID: AB_312691), CD8-Pacific Blue (clone 53-6.7, BioLegend; RRID: AB_493425), and CD19-PE/Cy7 (clone 1D3, BD; RRID:AB_10894021) or with CD45-APC/Cy7 (clone 30F11, BioLegend; RRID: AB_312981), CD3-BV605 (clone 17A2, BioLegend; RRID: AB_2562039), CD19-BV605 (clone 6D5, BioLegend; RRID: AB_11203538), Ly6G-BV711 (clone 1A8, BD; RRID:AB_2738520), F4/80-BV421 (clone T45-2342, BD; RRID: AB_2734779), NK1.1- PE/Cy7 (clone PK136, BioLegend; RRID: AB_389364), and CD11b-FITC (clone M1/70, BioLegend; RRID: AB_312789) in FACS buffer (PBS, 0.5% BSA, 0.02% NaN$_3$) supplemented with Fc-Block (True Stain anti-mouse CD16/32, clone 93; BioLegend; RRID: AB_1574975) for 30 min at 4°C in the dark, washed with PBS, and filtered and resuspended in PBS with 10 µM EDTA. CD4+ T cells (CD45+CD3+CD4+), CD8+ T cells (CD45+CD3+CD8+), B cells (CD45+CD19+), and macrophages (CD45+CD3-CD19-Ly6G-NK1.1-CD11b+ F4/80+) were sorted into collection tubes coated with FCS and filled with complete RPMI 1640 medium (1% penicillin, streptomycin, 0.1% 2-ME) with 20% FCS using a FACSAria device (BD Biosciences). Purity of sorted populations was routinely above 95%. Then cells were pelleted by centrifugation at 300 × g for 10 min at 4°C, dry frozen in liquid nitrogen, and stored at –80°C until RNA isolation. For isolation of microglia, mice were intracardially perfused with ice-cold PBS immediately after killing by inhalation of CO$_2$ to remove blood from intracranial vessels. Brain and spinal cord were excised with sterile instruments, and mechanically dissected and incubated with agitation in RPMI medium 1640 (PAN-Biotech) containing 1 mg/ml collagenase A (Roche) and 0.1

mg/ml DNaseI (Merck) for 60 min at 37°C. Tissue was triturated through a 40 µm cell strainer and washed with PBS (300 × *g*, 10 min, 4°C). Homogenized tissue was resuspended in 30% isotonic Percoll (GE Healthcare, Chicago, USA) and underlaid with 78% isotonic Percoll. After gradient centrifugation (1500 × *g*, 30 min, 4°C), CNS-immune cells were recovered from the gradient interphase and washed twice in ice-cold PBS. For fluorescence-activated cell sorting, cell suspensions were stained with Fixable Viability Stain 700, followed by staining of surface antigens with CD45-APC/Cy7 (clone 30F11, Biolegend; RRID: AB_312981), and CD11b-PerCPCy5.5 (clone M1/70, Biolegend; RRID: AB_312981) and Fc-Block as described above. Microglia (CD45med CD11b+) were sorted and processed for RNA isolation as described above.

## Quantitative real-time PCR

RNA was purified using the RNeasy Mini Kit (Qiagen) and reverse transcribed to cDNA with the RevertAid H Minus First Strand cDNA Synthesis Kit (Thermo Fisher) according to the manufacturer's instructions. Gene expression was analyzed by quantitative real-time PCR performed in an ABI Prism 7900 HT Fast Real-Time PCR System (Applied Biosystems, Waltham, USA) using TaqMan Gene Expression Assays (Thermo Fisher) for *Ppargc1a* (*Pgc-1α*; Mm00464452_m1), *Alas* (Mm01235914_m1), *Cox4i1* (Mm01250094_m1), *Cs* (Mm00466043_m1), *Nrf1* (Mm01135606_m1), *Tfam* (Mm00447485_m1), and *Tbp* (Mm01277042_m1). Gene expression was calculated as $2^{-\Delta CT}$ relative to *Tbp* as endogenous control.

## mtDNA copy number quantification

DNA was purified using the DNeasy Blood and Tissue Kit (Qiagen) according to the manufacturer's instructions. To destroy RNA, samples were treated with RNAse (Qiagen). Mitochondrial DNA copy numbers relative to diploid chromosomal DNA content were analyzed by quantitative real-time PCR performed in an ABI Prism 7900 HT Fast Real-Time PCR System (Applied Biosystems) using TaqMan Gene Expression Assays (Thermo Fisher) for *Cox2l* (Mm03294838_g1) and *β-actin* (Mm_00607939_s1). MtDNA copy numbers were quantified as $2^{-\Delta CT}$ relative to β-actin.

## *Ppargc1a* DNA copy number quantification

DNA was lysed with 50 µl QuickExtract DNA Extraction Solution (Lucigen, Middleton, USA). Copy numbers were analyzed by quantitative real-time PCR performed in an ABI Prism 7900 HT Fast Real-Time PCR System (Applied Biosystems) using TaqMan Copy Number Assays (Thermo Fisher) for *Ppargc1a* (*Pgc-1α*, Mm00164544_cn, FAM) and *Tfrc* (TaqMan Copy Number reference assay for mouse, VIC) in a duplex PCR. *Ppargc1a* copy numbers were quantified as $2^{-\Delta CT}$ relative to *Tfrc*.

## Immunoblot (Western Blot, WB)

After dissection, 15 mg of mouse spinal cord were lysed in 500 µl RIPA buffer supplemented with 50× protease inhibitor (complete, Sigma-Aldrich) and 2% 50x phosphatase (PhosSTOP, Sigma Aldrich), homogenized and incubated at 4°C for 30 min on a rotating wheel (GLW Storing Systems GmbH), and subsequently centrifuged at 3200 × *g* for 5 min at 4°C. Respective total protein amount of supernatant was determined with bicinchoninic acid protein assay. Only for validation of antibodies lambda protein phosphatase (New England BioLabs Inc, Ipswich, USA) was added to the samples. Samples were denatured by boiling 12 µg protein in a mixture of 15 µl 4× NuPAGE sample buffer (novex), 6 µl 10× Bolt sample reducing agent (novex), and RIPA buffer in a total volume of 60 µl for 5 min at 95°C. Samples were then loaded onto a Bolt 4–12% Bis-Tris Plus Gel (Invitrogen) in a Mini Gel Tank (Life Technologies, Carlsbad, USA). Chambers were filled with 1× Bolt MOPS SDS running buffer. Proteins were separated at 165 V for 1 hr. Spectra Multicolor High Range Protein Ladder (Thermo Fisher) was used as a protein standard. Proteins were then transferred to a nitrocellulose membrane at 10 V for 1 hr. For immunodetection, membrane was first blocked in 5% BSA in 1× TBS-T for 1 hr at room temperature and then incubated overnight at 4°C on a rotation wheel in 5% BSA in 1× TBS-T containing the primary antibodies against the following proteins: PGC-1α (rabbit 1:2000; Novus Biologicals; RRID: AB_1522118), phosphorylated PGC-1αS570 (rabbit 1:1000; R and D Systems, Minneapolis, USA; RRID: AB_10890391), and β-actin (rabbit 1:1000; Cell Signaling Technology, Cambridge, UK; RRID: AB_330288). The membrane was washed thrice with 1× TBS-T for 5 min each and subsequently incubated for 1 hr at RT on a shaker in 5% BSA in 1× TBS-T

supplemented with a horseradish-peroxidase conjugated secondary antibody (goat 1:15000; LI-COR Biosciences, Lincoln, USA; RRID: AB_2721264). The membrane was washed again three times in 1× TBS-T at RT for 5 min. Bound antibodies were detected by chemiluminescence using the Western-Sure ECL Substrate (LI-COR Biosciences) using the CCD camera-based ImageQuant LAS 4000 Mini system (GE Healthcare).

## Statistics

No statistical methods were used to predetermine sample sizes; our sample sizes are similar to those reported in previous publications (*Schattling et al., 2019*; *Schattling et al., 2012*). Experimental data were analyzed using Prism 8 software (GraphPad) and are presented as mean values ± s.e.m. Statistical analyses were performed using the appropriate test indicated in the figure legends. Outlier identification was first performed via Grubb's test with $p = 0.05$. Shapiro–Wilk test was used to analyze normality. In normally distributed data, differences between two experimental groups were determined by unpaired, two-tailed Student's t-test; in non-normally distributed data, we tested differences between two experimental groups by unpaired, two-tailed Mann–Whitney test. Differences between three experimental groups were determined by multiple comparisons test following one-way ANOVA. Significant results are indicated by asterisk: *$p < 0.05$, **$p < 0.01$, ***$p < 0.001$, ****$p < 0.0001$.

## Acknowledgements

We thank the UKE Mouse Pathology Facility for histopathology of EAE mice and the UKE Vector Facility for generating the calcium indicator viruses and the animal facility. We thank Hilmar Bading and Peter Bengtson for advice on calcium imaging. Funding SCR is supported by a Clinician-Scientist Fellowship from the Stifterverband für die Deutsche Wissenschaft and the Hertie Network of Excellence in Clinical Neuroscience of the Gemeinnützige Hertie-Stiftung. This work was supported by grants of the Deutsche Forschungsgemeinschaft (DFG) to MF (FR1638/3-1, FR1638/3-2) and MAF (FR1720/9-1, FR1720/9-2), which are part of the DFG FOR2289.

## Additional information

### Funding

| Funder | Grant reference number | Author |
| --- | --- | --- |
| Stifterverband | Clinician-Scientist Fellowship | Sina C Rosenkranz |
| Gemeinnützige Hertie-Stiftung | Hertie Network of Excellence in Clinical Neuroscience | Sina C Rosenkranz Jan Broder Engler |
| Deutsche Forschungsgemeinschaft | FR1638/3-1 | Marc Freichel |
| Deutsche Forschungsgemeinschaft | FR1638/3-2 | Marc Freichel |
| Deutsche Forschungsgemeinschaft | FR1720/9-1 | Manuel A Friese |
| Deutsche Forschungsgemeinschaft | FR1720/9-2 | Manuel A Friese |

The funders had no role in study design, data collection and interpretation, or the decision to submit the work for publication.

### Author contributions

Sina C Rosenkranz, Conceptualization, Data curation, Formal analysis, Supervision, Investigation, Visualization, Methodology, Writing - original draft; Artem A Shaposhnykov, Conceptualization, Data curation, Software, Investigation, Visualization, Methodology, Writing - review and editing; Simone Träger, Vanessa Roth, Vanessa Vieira, Nanne Paauw, Simone Bauer, Celina Schwencke-Westphal,

Charlotte Schubert, Lukas Can Bal, Investigation, Methodology, Writing - review and editing; Jan Broder Engler, Data curation, Software, Formal analysis, Visualization, Methodology, Writing - review and editing; Maarten E Witte, Data curation, Investigation, Methodology, Writing - review and editing; Benjamin Schattling, Conceptualization, Investigation, Methodology, Writing - review and editing; Ole Pless, Jack van Horssen, Conceptualization, Methodology, Writing - review and editing; Marc Freichel, Conceptualization, Resources, Formal analysis, Funding acquisition, Methodology, Writing - review and editing; Manuel A Friese, Conceptualization, Resources, Formal analysis, Supervision, Funding acquisition, Visualization, Methodology, Writing - original draft, Project administration

**Author ORCIDs**
Sina C Rosenkranz (iD) https://orcid.org/0000-0002-5228-4266
Artem A Shaposhnykov (iD) https://orcid.org/0000-0001-6772-6074
Jan Broder Engler (iD) https://orcid.org/0000-0002-3169-2076
Maarten E Witte (iD) https://orcid.org/0000-0002-1407-6220
Vanessa Vieira (iD) https://orcid.org/0000-0002-0205-9669
Celina Schwencke-Westphal (iD) https://orcid.org/0000-0002-5105-4290
Charlotte Schubert (iD) https://orcid.org/0000-0002-2967-4290
Lukas Can Bal (iD) https://orcid.org/0000-0003-4731-2311
Benjamin Schattling (iD) https://orcid.org/0000-0001-8809-9073
Ole Pless (iD) https://orcid.org/0000-0002-1468-316X
Jack van Horssen (iD) https://orcid.org/0000-0003-4078-7402
Marc Freichel (iD) http://orcid.org/0000-0003-1387-2636
Manuel A Friese (iD) https://orcid.org/0000-0001-6380-2420

**Ethics**

Animal experimentation: All animal care and experimental procedures were performed according to institutional guidelines and conformed to requirements of the German Animal Welfare Act. All animal experiments were approved by the local ethics committee (Behörde für Soziales, Familie, Gesundheit und Verbraucherschutz in Hamburg; G22/13 and 122/17). We conducted all procedures in accordance with the ARRIVE guidelines (Kilkenny et al., 2010).

**Decision letter and Author response**
Decision letter https://doi.org/10.7554/eLife.61798.sa1
Author response https://doi.org/10.7554/eLife.61798.sa2

# Additional files

**Supplementary files**
• Transparent reporting form

**Data availability**

All current data have been provided. All individual data are reported within the figure graphs as scatter plots. Numerical data for Figure 3E, 5A and 5C can be found as Source Data. Enrichment analysis was performed using the function 'gseGO' of the R package clusterProfiler; plotting was performed with the R packages ggplot2, clusterProfiler and tidyheatmaps. Normalization, calcium spikes, base line detection, and analysis of the calcium clearance rate were performed with the custom-made script written on Python 3.6 (https://github.com/scriptcalcium/PGC1alpha; copy archived at https://archive.softwareheritage.org/swh:1:rev:af3c206a43f8b6e9fdbd7707c9b4601287d5968b/).

The following previously published dataset was used:

| Author(s) | Year | Dataset title | Dataset URL | Database and Identifier |
|---|---|---|---|---|
| Schattling B, Engler | 2019 | Bassoon proteinopathy drives | https://www.ncbi.nlm. | Gene Expression |

| JB, Volkmann C, Rothammer N, Woo MS, Petersen M, Winkler I, Kaufmann M, Rosenkranz SC, Fejtova A, Thomas U, Bose A, Bauer S, Träger S, Miller KK, Brück W, Duncan KE, Salinas G, Soba P, Gundelfinger ED, Merkler D, Friese MA | neurodegeneration in multiple sclerosis | nih.gov/geo/query/acc.cgi?acc=GSE104899 | Omnibus, GSE104899 |

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
