## [Decision Letter]

**Acceptance summary:**

The study by Rosenkranz and colleagues explores the relationship between mitochondrial activity and neurodegeneration associated with inflammatory lesions in the central nervous system, a scientifically and clinically important topic with direct relevance to multiple sclerosis (MS). Their results reinforce the therapeutic potential of the transcription factor PGC1α for treating neuroinflammation associated with neurodegeneration, specifically in multiple sclerosis.

**Decision letter after peer review:**

Thank you for submitting your article "Enhancing mitochondrial activity in neurons protects against neurodegeneration in CNS inflammation" for consideration by *eLife*. Your article has been reviewed by three peer reviewers, one of whom is a member of our Board of Reviewing Editors, and the evaluation has been overseen by Satyajit Rath as the Senior Editor. The reviewers have opted to remain anonymous.

The reviewers have discussed the reviews with one another and the Reviewing Editor has drafted this decision to help you prepare a revised submission.

Summary:

The study by Rosenkranz and colleagues explores the relationship between mitochondrial activity and neurodegeneration in inflammatory CNS lesions, a scientifically and clinically important topic with direct relevance to multiple sclerosis. They found transcription of mitochondrial genes related to the electron transport chain (ETC) are decreased in motor neurons derived from EAE animals in line with a reduced complex IV activity that is observed throughout the spinal gray matter in EAE. They link these changes to the transcription factor PGC1α. Manipulating PGC1α levels modulated mitochondrial function and the pathological features of an EAE model in a manner consistent with PGC1α having therapeutic potential for neuroinflammation associated with neurodegeneration, specifically in multiple sclerosis.

Essential revisions:

1) The majority of lesions in most EAE models are located lumbar spinal cord and the clinical score indicates that this is also the case in this paper. However, the entire analysis was performed in the cervical spinal cord. As a result of this, the changes observed would be mostly distal to the most heavily inflamed areas of the spinal cord. It would be informative and important for the interpretation of data to know whether the key findings of the study and in particular the beneficial effects of Pgc -1α overexpression on complex IV activity and neuronal survival (in the absence of an effect on the local inflammatory reaction) are also present in the lumbar spinal cord.

2) The identification of Pgc1α as a therapeutic target is derived from the reduced expression of Pgc-1α-dependent ETC genes in the EAE spinal cord (Figure 2). If possible, it would be helpful to show that (at least some of) these changes to neuronal gene expression are reversed by Pgc-1α overexpression in the EAE spinal cord and provide further support for PGC1α as a therapeutic approach to MS.

3) NeuN is used a marker for visualizing neurons, but the expression of NeuN can be modulated, particularly under inflammatory conditions. An additional reference marker should be considered before concluding differences between EAE and control animals.

4) Both conditional Pgc-1α deficient and Pgc-1α overexpressing mice show an altered density of neurons in the GM and the ventral spinal cord in the context of EAE, confirm the neuronal densities in these areas do not differ in the corresponding age-matched healthy mice.

5) In Figure 3E, the images are labeled as "representative" but do not appear to represent the data from hPNC. A quick analysis of the images suggests 5-8 fold increase (not the 1.5 -2.5 indicated by the MFI). Are images from hPNC or cPNC? In the graph, it appears there may be a bimodal distribution of cells for hPNC from PGC1α group (possibly cPNC as well), is there any technical reason that might account for clustering? For cPNC, it appears there are a few outliers. Why was t-test used not ANOVA?

6) The authors claim that there is no difference in immune response between wild type and PGC-1αTT mice, however in Figure 5D, the micrograph for IBA does not support this claim, and appears to contradict the graph. There is clearly more IBA in the chosen tissue section from the PGC-1αTT mice, compared to wild type, but the graph suggests a decrease (although does not appear to be significant).

[Editors' note: further revisions were suggested prior to acceptance, as described below.]

Thank you for resubmitting your work entitled "Enhancing mitochondrial activity in neurons protects against neurodegeneration in a mouse model of multiple sclerosis" for further consideration by *eLife*. Your revised article has been evaluated by three peer reviewers, one of whom is a member of our Board of Reviewing Editors, and the evaluation has been overseen by Satyajit Rath as the Senior Editor.

Essential revision:

The manuscript has been improved but there is a remaining issue that need to be addressed before acceptance, as outlined below:

The authors address the appropriateness of the model by referring to a study showing that clinical scores correlate better with cervical spinal cord lesions in EAE, however, they did not incorporate this reference nor address the concern of approach in the revised manuscript. Given this was part of essential revision, the authors should consider addressing choice of focusing on the cervical region of SC rather than lumbar in the manuscript proper.

Other concerns are adequately addressed.

---

## [Author Response]

Essential revisions:1) The majority of lesions in most EAE models are located lumbar spinal cord and the clinical score indicates that this is also the case in this paper. However, the entire analysis was performed in the cervical spinal cord. As a result of this, the changes observed would be mostly distal to the most heavily inflamed areas of the spinal cord. It would be informative and important for the interpretation of data to know whether the key findings of the study and in particular the beneficial effects of Pgc -1α overexpression on complex IV activity and neuronal survival (in the absence of an effect on the local inflammatory reaction) are also present in the lumbar spinal cord.

We thank the reviewers for their comment and welcome the opportunity to explain our rationale by first drawing their attention to previous findings, in which it was shown that clinical scores correlate better with cervical spinal cord lesions in EAE (Philippe et al., 2017). Moreover, in our own previous studies (Friese et al., 2007; Schattling et al., 2012, 2019; Rosenkranz et al., 2020) we characterized immune cell infiltration, demyelination and neuronal loss in the cervical spinal cord that also showed robust correlation with the clinical phenotype. While we agree that the lesions in EAE are similarly present in the lumbar spinal cord, out of the above reasons we believe that our data sufficiently back our interpretations. We hope that the reviewers regard our approach as appropriate.

2) The identification of Pgc1α as a therapeutic target is derived from the reduced expression of Pgc-1α-dependent ETC genes in the EAE spinal cord (Figure 2). If possible, it would be helpful to show that (at least some of) these changes to neuronal gene expression are reversed by Pgc-1α overexpression in the EAE spinal cord and provide further support for PGC1α as a therapeutic approach to MS.

We thank the reviewer for pointing this out and agree that this would be an interesting analysis. Thus, we performed a qPCR analysis of the PGC-1α-regulated ETC genes, which we previously detected to be downregulated in EAE, in wild-type and *Thy1*-Ppargc1a primary hippocampal neurons. Importantly, we could observe a significant increase of 11 out of 17 analyzed genes (new Figure 3A), supporting a therapeutic approach. We agree that analysis of these genes in neurons during EAE would have been a nice addition, however due to Corona restrictions we were limited in our breeding capacity and could not cross our *Chat-L10-eGFP*Bactrap mice to *Ppargc1a*-overexpressing animals. However, we believe that our new data set sufficiently supports our claims.

3) NeuN is used a marker for visualizing neurons, but the expression of NeuN can be modulated, particularly under inflammatory conditions. An additional reference marker should be considered before concluding differences between EAE and control animals.

We agree with the reviewer that NeuN staining can be diminished in EAE, thereby compromising our analysis. Following the reviewer’s suggestion, we performed stainings with HuC/HuD, an alternative neuronal marker that now served us as reference for normalization of neuronal counts. By doing so, we revealed that the reduction of COX activity at day 30 was partly driven by neuronal loss, which is now shown in the new Figure 1G. Furthermore, we added an analysis of the EAE tissue from *Thy1*-Ppargc1a vs. wild-type and *Ppargc1a*^flx/flx^ × *Eno2*^cre+^ vs. *Ppargc1a*^flx/flx^ at day 40 post immunization by using HuC/HuD staining as neuronal marker. In comparison to the NeuN staining we detected higher overall neuronal numbers with the HuC/HuD staining in all tissues, however the significant differences between *Thy1*-Ppargc1a vs. wild-type and *Ppargc1a*^flx/flx^ × *Eno2*^cre+^ vs. *Ppargc1a*^flx/flx^ were not affected by using the different stainings (new Figure 5—figure supplement 1G and I).

4) Both conditional Pgc-1α deficient and Pgc-1α overexpressing mice show an altered density of neurons in the GM and the ventral spinal cord in the context of EAE, confirm the neuronal densities in these areas do not differ in the corresponding age-matched healthy mice.

We agree that this is an important control as the initial numbers of neurons could influence our EAE outcome measures. Therefore, we performed stainings and analyses of healthy *Ppargc1a*^flx/flx^ × *Eno2*^cre+^ vs. *Ppargc1a*^flx/flx^ and *Thy1*-Ppargc1a vs. wild-type animals and could confirm that the different genotypes did not differ in neuronal counts in the GM or VH. This is now shown in the new Figure 5—figure supplement 1H and J.

5) In Figure 3E, the images are labeled as "representative" but do not appear to represent the data from hPNC. A quick analysis of the images suggests 5-8 fold increase (not the 1.5 -2.5 indicated by the MFI). Are images from hPNC or cPNC? In the graph, it appears there may be a bimodal distribution of cells for hPNC from PGC1α group (possibly cPNC as well), is there any technical reason that might account for clustering? For cPNC, it appears there are a few outliers. Why was t-test used not ANOVA?

We thank the reviewer for this advice and replaced the representative image of the WT hPNC (new Figure 3F). As *Thy1* expression shows heterogeneity this might be the explanation of the detected range of TMRE in *Thy1*-Ppargc1a mice. However, as indicated in the statistical methods we performed an outlier analysis and none of the data points shown were detected as outlier in our analysis. T-tests were used to compare the two groups of interest with each other – *Thy1*-Ppargc1a vs. wild-type.

6) The authors claim that there is no difference in immune response between wild type and PGC-1αTT mice, however in Figure 5D, the micrograph for IBA does not support this claim, and appears to contradict the graph. There is clearly more IBA in the chosen tissue section from the PGC-1αTT mice, compared to wild type, but the graph suggests a decrease (although does not appear to be significant).

We agree that the figure was not ideally chosen and replaced the representative image of the Iba1 stainings in Figure 5D.

References:

Philippe A, Quenault A, Martinez S, Lizarrondo D, Gauberti M, Defer G (2017) Prediction of disease activity in models of multiple sclerosis by molecular magnetic resonance imaging of P-selectin. PNAS 114:6116–6121.Rosenkranz SC, Shaposhnykov A, Schnapauff O, Epping L, Vieira V, Heidermann K, Schattling B, Tsvilovskyy V, Liedtke W, Meuth SG, Freichel M, Gelderblom M, Friese MA (2020) TRPV4-Mediated Regulation of the Blood Brain Barrier Is Abolished During Inflammation. Front Cell Dev Bio 8:1–9.

[Editors' note: further revisions were suggested prior to acceptance, as described below.]

Essential revision:The manuscript has been improved but there is a remaining issue that need to be addressed before acceptance, as outlined below:The authors address the appropriateness of the model by referring to a study showing that clinical scores correlate better with cervical spinal cord lesions in EAE, however, they did not incorporate this reference nor address the concern of approach in the revised manuscript. Given this was part of essential revision, the authors should consider addressing choice of focusing on the cervical region of SC rather than lumbar in the manuscript proper.

We thank the reviewers for their positive and encouraging feedback and apologize for the oversight not addressing this point in our manuscript. We now included the following sentence in the Materials and methods section and cited the respective reference: “Cervical spinal cord was used as it was shown that clinical scores correlate better with cervical than lumbar spinal cord lesions in EAE (Fournier et al., 2017).”